# Aerosol monitoring in Siberia using an 808 nm automatic compact lidar

Gerard Ancellet[1], Iogannes E. Penner[2], Jacques Pelon[1], Vincent Mariage[1], Antonin Zabukovec[1], Jean Christophe Raut[1], Grigorii Kokhanenko[2], and Yuri S. Balin[2]

[1]LATMOS/IPSL, Sorbonne Université, CNRS, UVSQ, Paris, France
[2]Zuev Institute of ATmospheric Optics, Russian Academy of Sciences, Tomsk, Russia

**Correspondence:** Gerard Ancellet : gerard.ancellet@upmc.fr

**Abstract.** Our study is providing new information on aerosol type seasonal variability and sources in Siberia using observations (ground-based lidar and sun-photometer combined with satellite measurements). A micropulse lidar emitting at 808 nm provided almost continuous aerosol backscatter measurements for 18 months (April 2015 to September 2016) in Siberia, near the city of Tomsk ($56^o$N, $85^o$E). A total of 540 vertical profiles (300 daytime and 240 nighttime) of backscatter ratio and aerosol extinction have been retrieved over periods of 30 min, after a careful calibration factor analysis. Lidar ratio and extinction profiles are constrained with sun-photometer Aerosol Optical Depth at 808 nm ($AOD_{808}$) for 70% of the daytime lidar measurements, while 26% of the nighttime lidar ratio and $AOD_{808}$ greater than 0.04 are constrained by direct lidar measurements at an altitude greater than 7.5 km and where a low aerosol concentration is found. An aerosol source apportionment using the Lagrangian FLEXPART model is used in order to determine the lidar ratio of the remaining 48% of the lidar database. Backscatter ratio vertical profile, aerosol type and $AOD_{808}$ derived from micropulse lidar data are compared with sunphotometer $AOD_{808}$ and satellite observations (CALIOP spaceborne lidar backscatter and extinction profiles, Moderate Resolution Imaging Spectroradiometer (MODIS) $AOD_{550}$ and Infrared Atmospheric Sounding Interferometer (IASI) CO column) for three case studies corresponding to the main aerosol sources with $AOD_{808} > 0.2$ in Siberia. Aerosol typing using the FLEXPART model is consistent with the detailed analysis of the three case studies. According to the analysis of aerosol sources, the occurrence of layers linked to natural emissions (vegetation, forest fires and dust) is high (56%), but anthropogenic emissions still contribute to 44% of the detected layers (1/3 from flaring and 2/3 from urban emissions). The frequency of dust events is very low (5%). When only looking at $AOD_{808} > 0.1$, contributions from Taiga emissions, forest fires and urban pollution become equivalent (25%), while those from flaring and dust are lower (10%-13%). The lidar data can also be used to assess the contribution of different altitude ranges to the large AOD. For example, aerosols related to the urban and flaring emissions remain confined below 2.5 km, while aerosols from dust events are mainly observed above 2.5 km. Aerosols from forest fire emissions are on the opposite observed both within and above the Planetary Boundary Layer (PBL).

# 1 Introduction

Knowledge about the aerosol particles distribution and properties has been identified by the Intergovernmental Panel on Climate Change (IPCC) as an important source of uncertainty in climate change (Stocker et al., 2013). Siberia represents 10% of land surface and 30% of forested surfaces globally and plays a key role in the Earth system. Parts of the Siberian Arctic are warming at some of the strongest rates on Earth (2 K/50 yrs) (Stocker et al., 2013). Increased resource extraction and opening of the Northern Sea Route are leading to new sources of pollution. A recent Arctic Council report identified aerosols from Asian pollution and from gas flaring associated with oil/gas production in northern Siberia as key sources (AMAP, 2015). The impact of pollutants in Siberia is underestimated likely because of poor knowledge of Russian emissions (Huang et al., 2015; Bond et al., 2013), and poor process and feedback representation in climate models (Eckhardt et al., 2015; Arnold et al., 2016).

Radiative effects are highly dependent on the vertical stratification of aerosols. Clear-sky longwave forcing and cloudy-sky shortwave forcing of dust layer are very sensitive to the layer altitude, while the sign of the radiative effect of a biomass burning smoke layer depends on the presence of underlying stratus (Mishra et al., 2015; Tosca et al., 2017). Ground-based and spaceborne lidar observations are now key elements of aerosol monitoring because they can provide regular observations. The analysis of data from the European Aerosol Lidar Network (EARLINET) has significantly improved our knowledge of aerosol sources and long-range transport in Europe (Pappalardo et al., 2014). This has been mostly achieved benefiting from the extended implementation of Raman lidar systems, e.g. in Mattis et al. (2004); Ansmann et al. (2001). However other solutions such as micropulse lidars or improved ceilometers have been identified that may significantly contribute to improve our knowledge on aerosol properties provided consolidated approaches are developed (Pelon et al., 2008; Wiegner et al., 2014; Mariage et al., 2017). Aerosol backscatter and extinction profiles using such systems have been started from NASA's Micropulse Lidar Network (MPLNET) in North America and Asia (Campbell et al., 2002; Misra et al., 2012). Such systems have limited range capabilities during daytime, when sun-photometer observations are available, but the advantages are their low cost and their simple operation mode. Micropulse lidars have been operated at various wavelengths in the visible and near-infrared, none in the UV mostly because eye-safety is guaranteed by low pulse energy emission. Identified constraints are then to avoid strong water vapor bands in the near infrared, and retrieve molecular scattering that can be used as a reference for calibration, e.g. systems operating at 1064 nm have provided valuable information on aerosol (Wiegner and Geiß, 2012; Wiegner et al., 2014).

The Commonwealth of Independent States lidar network (CIS Linet) has also been established in Belarus, Russia and the Kyrgyzstan Republic (Chaikovsky et al., 2006), mostly with backscatter lidars, but very few analyses of regular lidar observations have been published for Siberia. The main contribution is the analysis of 84 multi-wavelength lidar observations from March 2006 to October 2007 in Samoilova et al. (2010) showing different optical properties of aerosols for the cold and warm season in Tomsk, Russia. The spectral variation of the lidar ratio in the boundary layer is also consistent with the optical properties of an urban aerosol model (Samoilova et al., 2012). Another comprehensive study on the vertical distribution of aerosols in Russia comes from a summer field campaign with a mobile lidar in June 2013 making a road transect between

Smolensk (32ºE, 54ºN) and Lake Baikal (107ºE, 51ºN) (Dieudonné et al., 2015). The dust outbreak (close to 70ºE) and the biomass burning have been identified as the main aerosol sources during this campaign.

The constellation of satellites grouped in A-Train provides active and passive measurements of the optical properties of aerosols and clouds. The primary optical properties of aerosols derived from passive instrument measurements such as Moderate Resolution Imaging Spectroradiometer (MODIS) on TERRA and AQUA platforms under clear sky conditions are the aerosol optical depth (AOD) and Angström exponent (AE), which is a parameter indicative of particle size (Levy et al., 2013). The Multi-angle Imaging SpectroRadiometer (MISR) instrument provides similar parameters with a better accuracy on AE (Kahn and Gaitley, 2015). Forest fires or gas flaring emissions are also derived during the night from the Visible Infrared Imaging Radiometer Suite (VIIRS) (Schroeder et al., 2014) The Cloud-Aerosol Lidar and Infrared Pathfinder Satellite Observation (CALIPSO) mission (Winker et al., 2009) has proven very useful in characterizing cloud and aerosol distribution on a global scale (Winker et al., 2013). The level 2 products of Cloud-Aerosol Lidar with Orthogonal Polarization (CALIOP), namely the 5-km aerosol layer products (AL2) allow the indirect calculation of vertical profiles of extinction and of AOD (Omar et al., 2009; Young and Vaughan, 2009). The observations made by the CALIOP lidar provide the optical properties of the aerosol layers at two different wavelengths (532 nm, 1064 nm) and the depolarization ratio can be calculated using parallel and perpendicular backscatter signals at 532 nm measured by two orthogonal polarized channels. Regional aerosol distribution studies have been conducted for the high latitudes of the northern hemisphere (Di Pierro et al., 2013), for European Arctic (Ancellet et al., 2014; Law et al., 2014) and for the Arctic ice sheet (Di Biagio et al., 2018), but there are no similar studies for central Siberia.

In this paper we report on measurements made day and night during 18 months by a micro lidar at 808 nm located near the city of Tomsk, Russia (56ºN, 85ºE). Quantitative retrievals using micropulse lidar systems such as proposed here imply a proper calibration (Mariage et al., 2017) and or the use of atmospheric references such as sun-photometers (Marenco et al., 1997; Welton et al., 2002; Pelon et al., 2008). In this study, we refine the analysis method to control nighttime calibration over a long time series, and extend it to daytime observations. A control of the performance is achieved through comparisons of AODs directly derived from micropulse lidar measurements with sun-photometer ones. This last parameter can be compared with the measurements of the CIMEL Electronique CE 318 sun-photometer, which is a part of AErosol RObotic NETwork (AERONET), (Holben et al., 1998) and located on the same site. The objective is then to use the micropulse lidar database to characterize the sources of aerosols that can be transported over the measurement site and to verify how they contribute to the vertical distribution of aerosols and to the optical thickness of the atmospheric column. Analysis of satellite observations (CALIPSO, MODIS, VIIRS, ...) measurements provide additional information on aerosol source variability and aerosol plume transport processes. The lidar system, signal processing and AOD retrieval method are described in section 2, while the section 3 presents the aerosol transport model and the aerosol sources. section 4 described the results of AOD retieval using the lidar calibrated signal, AERONET sun-photometer data and aerosol type from section 3. The results about the aerosol layer distribution are described and discussed in sections 5, 6.

## 2 Lidar data analysis

An eye-safe CIMEL CE372 lidar was installed in Tomsk in April 2015 to obtain continuous measurements of cloud and aerosol backscatter vertical profiles. The lidar was first installed on the roof of the Institute of Atmospheric Optics (IAO) for 4 months (April 2015-August 2015) before being moved in a thermostatically controlled box at Fonovaya Observatory, 50 km West of Tomsk (September 2015 to August 2016). It was then re-installed on the IAO roof for one month in September 2016 before being shut down for several months of maintenance. The lidar was installed near the local AERONET sun-photometer to obtain an independent measurement of the total AOD. This is necessary when no proper calibration can be applied nor molecular scattering identified above aerosol layers (Welton et al., 2000; Cuesta et al., 2008; Chaikovsky et al., 2016). This article will therefore focus on the analysis of the measurements collected over the period April 2015 to September 2016. In this section, the lidar will be described and the calibration method necessary to improve the retrieval of the AOD is presented. The methodology for the AOD retrieval is then described in section 2.3.

### 2.1 Lidar system description

The CIMEL CE372 lidar belongs to a new generation of lidar derived from the previous CE370 model operating in the visible (Dieudonné et al., 2013) and from the one specially developed for the IAOOS project (Mariage et al., 2017). The CE372 is a single wavelength system using a laser diode emitting 200 ns pulses at 808 nm, whose temperature is regulated by a Peltier device. The maximum output power is 18 mW with a repetition rate of 4.72 kHz (3.8 $\mu$J energy). The energy of the laser diode is recorded continuously with a photodiode and a 30 nm filter centered at 808 nm, but the energy measurement was only reliable during the night because the background solar radiation is still too high on the photodiode to make daytime measurements possible. The optical receiver includes a 10 cm diameter lens and a 0.6 nm filter to reduce background light. The detection unit is based on an Avalanche photodiode (APD) used in Geiger mode (Single Photon Counting Module from EXCELITAS) and a standard high-speed sampling and averaging electronic card from Cimel Electronique. The photocounting signal is delivered by the SPCM with a maximum frequency around 35 MHz and detection gate of 100 ns (15 m vertical resolution). Lidar profiles are recorded with an integration time of 1 min. The signal is corrected from saturation due to APD detector dead time (22 ns) using the methodology of Mariage et al. (2017). The background correction uses the average signal recorded between 20 and 30 km.

For each day 3 periods of 30 min are selected between 0 UT-12 UT (day), 12UT-20 UT (night), 20UT-24 UT (day) for the analysis of vertical aerosol profiles. The selection of the best interval of 30 min to average the 1-min lidar profiles is based on the elimination of very cloudy profiles. Data filtering with lidar radiometric detection of a cloud (day only), with search for layers showing very strong backscatter below 5 km (day and night) and for high opacity of the 0-3 km atmospheric layer (day and night), selects very efficiently the 30-min time periods with no cloud layers below 5 km. The following criteria are then applied to eliminate the lidar data considered too cloudy: all the profiles with a daytime sky level (SB) greater than 7000 counts/s or with a 150 m layer where the backscatter ratio is greater than 17 between 0 and 4.5 km, or with attenuated backscatter smaller than $10^{-4}$ km$^{-1}$sr$^{-1}$ between 3 and 8 km cloud layers below 5 km.

A total of 540 averaged profiles are thus available for aerosol profile analysis over the period April 2015 to September 2016 with 300 daytime profiles and 240 nighttime profiles.

An example of the attenuated backscatter vertical profile for a 30 min nighttime and daytime averaging in June 2015 is shown in Fig. 1. The signal is normalized to the molecular attenuated backscatter during the night at 9 km below a cirrus cloud observed above 10.5 km. The signal to noise ratio SNR is good enough to detect aerosol layers up to the tropopause during the night. Only aerosols below 3 km are detected during the day and the molecular reference signal cannot be accurately measured during the day.

As the alignment of the lidar remains very stable over time, the geometric overlap factor, OF, between the laser and the receiver is estimated between the surface and 500 m by averaging the profiles with mean attenuated backscatter ratio < 1.1 at 500 m and by assuming a constant scattering ratio between the surface and 500 m. This provides a sufficiently accurate geometric overlap factor to correct for the underestimation of the contribution of this altitude domain to the AOD assessment (Fig. 2) between 100 m and 500 m. Below 100 m, OF retrieved with this method is not accurate enough and we will assume a constant backscatter ratio between the surface and 100 m. This assumption induces a 2% error on the AOD assuming a constant extinction layer 1000 m deep and a 20% error on the scattering ratio below 100 m.

## 2.2 Lidar calibration

Owing to the low SNR of daytime lidar signal above 2-3 km altitude and the difficulty to always find an altitude zone where aerosol backscatter is negligible compared to molecular backscatter, we propose a specific methodology to determine the evolution of the lidar calibration factor. Indeed a precise calibration of the lidar first allows the determination of the daytime integrated backscatter assuming very low variation of calibration factor during the day. Daytime integrated backscatter is then used to derive the integrated lidar ratio using independent AOD measurement from a sun-photometer. During the night, calibrated lidar measurements are also useful to reduce the uncertainty on the calculation of the extinction profile to the relative error on the range corrected signal (PR2) and to that on the determination of the lidar ratio (Appendix A).

A first guess of the calibration coefficient K is obtained from a normalization of the minimum backscatter ratio R to 1 at an altitude between 4 km and 9 km for night profiles. The vertical profile of molecular backscatter is estimated from the pressure and temperature profiles after temporal and spatial interpolation of the 4 daily ERA-Interim ECMWF meteorological fields at 0.75°(Dee et al., 2011). The lidar backscatter ratio is averaged over 150 m to reduce the uncertainty on the molecular signal below 2.5% i.e. 3.2 times less than the 8% signal standard deviation shown in Fig. 1 during the night at 9 km. A first guess of the aerosol two-way transmittance $T_a^2$ between the altitude 100 m and the reference altitude $z_r$, chosen for normalization to the backscatter profile, is then calculated after determining the extinction profile with an a priori lidar ratio and using the backward inversion method described in Appendix A. An a priori value of 60 sr is chosen for the vertically averaged lidar ratio S at 808 nm because it corresponds to biomass burning or pollution aerosols using the lidar ratio look up table at 532 nm of the CALIPSO mission aerosol climatology (Omar et al., 2009) and the spectral variability of the lidar ratio between 500 nm and 808 nm proposed by Cattrall et al. (2005).

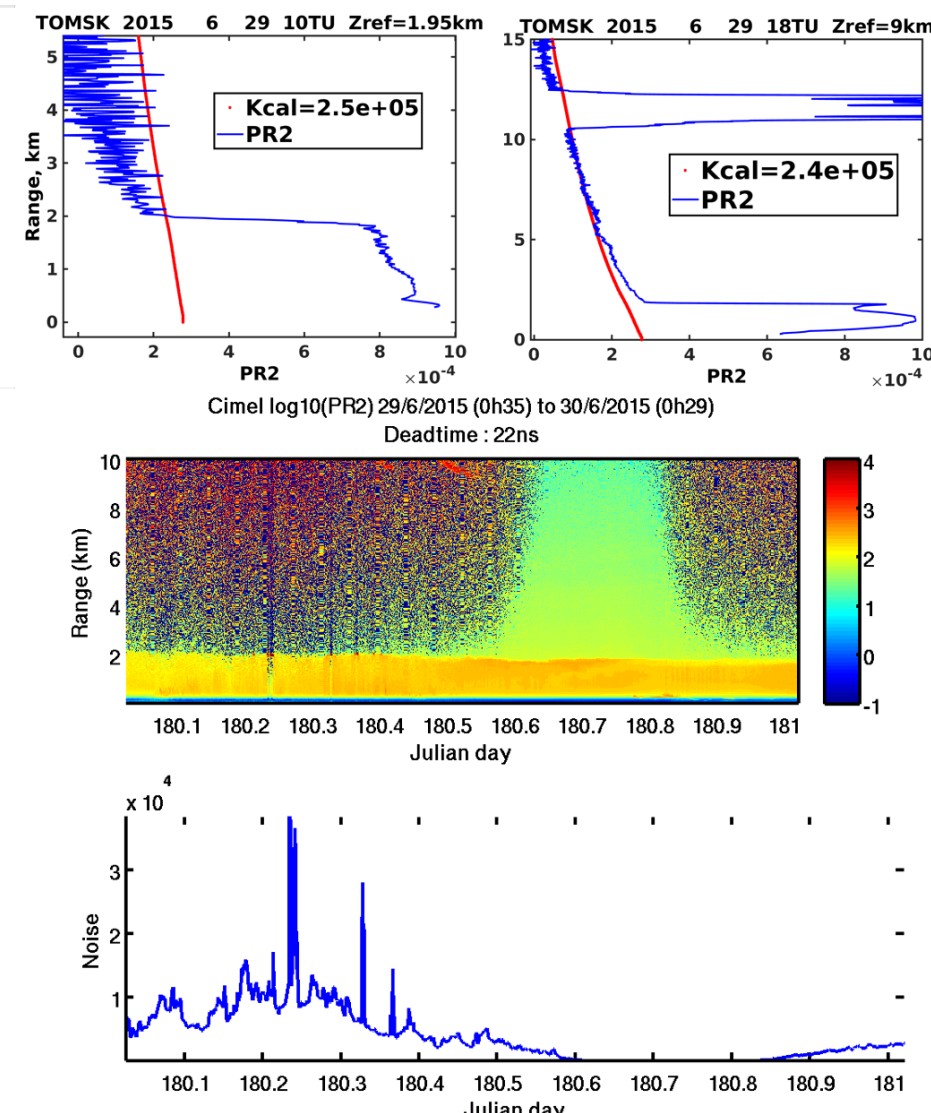

**Figure 1.** Vertical profiles of the attenuated backscatter signal (PR2) for daytime (top left) and nighttime (top right) averaged over 30 minutes on 29/06/2015 using a calibration constant K to normalize the nighttime PR2 to attenuated molecular backscatter at 10 km below the cirrus layer. The red curve is the attenuated molecular backscatter signal. Daily evolution of the vertical profiles of the log10 of the attenuated backscatter (central panel) and the background signal due to solar radiation (lower panel).

The second step in our estimation of lidar calibration is to select the nighttime profiles with two additional criteria: $z_r > 7.5$ km and $T_a^2 > 0.89$. There are 106 such profiles out of 540. This selection increases the probability of having a normalization zone with a good signal to noise ratio and a negligible contribution of particle backscatter ($z_r > 7.5$ km) and minimizing the normalization error due to an error on $T_a^2$ if S is very different from 60 sr ($T_a^2 > 0.89$). This corresponds to the profile selection

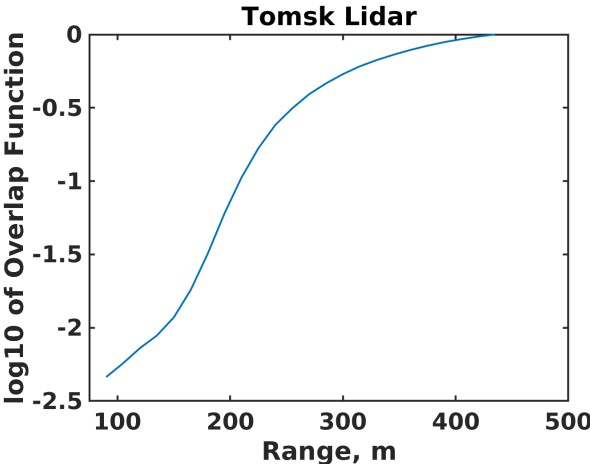

**Figure 2.** Geometrical overlap function used for the correction of the lidar data in log10 scale between 100 m and 500 m.

shown in the left part of figure 4. For these profiles, the calibration factors deduced from a normalization of the PR2 at $z_r$ are the best proxies for the lidar calibration. The corresponding optimal values $K_{opt}$ of the calibration factor are shown in Fig. 3 (black crosses). Since the error on the backscatter ratio at $z_r$ is below 3%, the error $\Delta K$ on this calibration factor depends mainly on the lidar ratio error $\Delta S$:

$$\frac{\Delta K}{K} = Ln(T_a^2)\frac{\Delta S}{S} \quad\quad\quad (1)$$

Assuming a 35% relative uncertainty on S (i.e. expected lidar ratio in the range 35-60 sr assuming that all the aerosol types can be encountered except the clean marine or dusty marine type (Omar et al., 2009)) and the Cattrall et al. (2005) S spectral variability, the error on $K_{opt}$ is less than 4% according to Eq. 1 when $T_a^2 > 0.89$.

The third step is to replace the calibration factors K for non-optimal conditions (daytime profiles on one hand and nighttime
with either AOD > 0.06 or clouds between 4 and 7.5 km) by interpolated values between the nearest $K_{opt}$ values (see the four arrows labeled with $K_{opt}$ in Fig. 4). If there are more than 10 days between two optimal calibration factors, the nearest value of $K_{opt}$ is chosen. If the interpolated value is greater than 20% of the calibration factor first guess divided by $T_a^2$, the latter is retained to take into account exceptionally lower optical transmission of the lidar (window icing, de-tuned filter) or a transient decrease in the emitted energy. Indeed the use of the interpolated calibration factor would lead to backscatter ratio much too
low in the free troposphere ($< 0.8.T_a^2$). There are less than 20 such cases between December 2015 and June 2016, therefore less than 3% of the cases studied have unusually low calibration factor.

The time evolution of K shown in Fig. 3 shows that the overall transmission of the lidar system increased by 30% when it was installed in the Fonovaya container in September 2015 and decreased again when it was operated again on the roof of the IAO for one month in September 2016. At the Fonovaya site the short-term variability ($< 10$ days) is much higher ($>$
15%) than at the Tomsk site where on the other hand the calibration constant increases regularly by 30% over 4 months. The

short-term variability is mainly related to changes in the optical transmission of the air-conditioned container window while the drift over 4 months with the initial conditioning of the CE372 on the roof of IAO is due to an improvement in the filter transmission at 808 nm during a gradual increase in outside temperatures. Analysis of the nighttime energy measurements does not indicate any significant variation in the energy emitted by the laser diode (<15%). To estimate our error on K values for non-optimal conditions (red points in Fig. 3), a good proxy is the difference between two optimal calibration factors derived for two observations made with a time difference < 1 day. Changes of $K_{opt}$ for such a short time period cannot be expected when aiming at calibration of daytime observations with a nighttime calibrated profiles. There are 23 pairs of $K_{opt}$ values with a 1-day time difference and the standard deviation of their difference, $\Delta K_{opt}$, is 2.5 $10^4$. Such a variability is then a limiting factor in our ability to calibrate the lidar for daytime observations or nighttime conditions with AOD>0.06 or clouds between 4 and 7.5 km. The corresponding accuracy on the calibration factor K is then of the order of 8% (2.5 $10^4$ / 3 $10^5$).

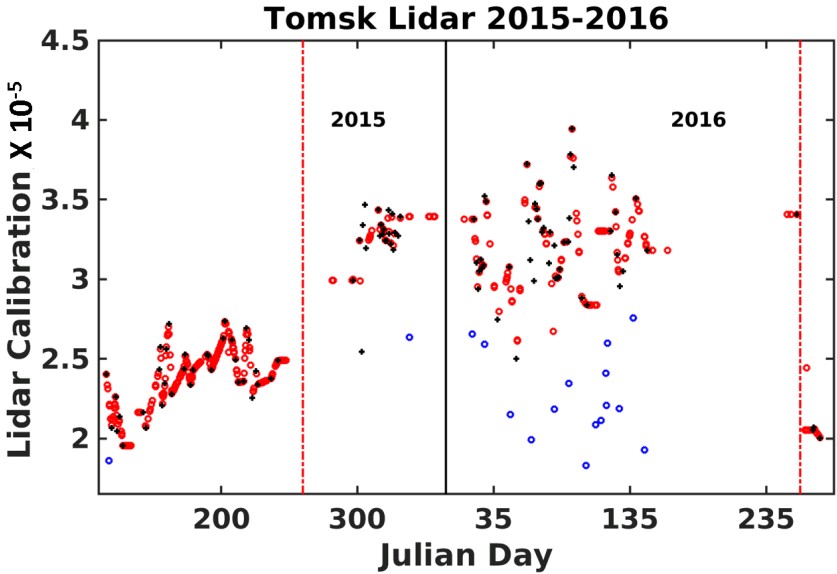

**Figure 3.** Time evolution from April 2015 to September 2016 of the lidar calibration factor (multiplied by $10^{-5}$). Dotted red lines correspond to major changes in lidar housing and expected change in calibration. Black crosses are for nocturnal profiles with molecular normalization at $z_r > 7$ km and aerosol two-way transmittance $T_a^2 >0.89$ (106 values out of 540), red dots are for calibration interpolated from optimal conditions, blue dots are for the few cases (20 out of 540) when calibration cannot be interpolated from optimal conditions.

## 2.3 Methodology for the lidar Aerosol Optical Depth (AOD) retrieval

Daytime indirect aerosol optical depth retrieval from the calibrated PR2 is based on the well known backward inversion of PR2 (Fernald, 1984; Klett, 1981) described in Appendix A, provided that independent measurement of $T_a^2$ is available to constrain the lidar ratio, e.g. using a sun-photometer (Chaikovsky et al., 2016; Cuesta et al., 2008). This work proposes a methodology for the AOD and backscatter ratio profile retrieval taking into account the different observation conditions described in Fig.

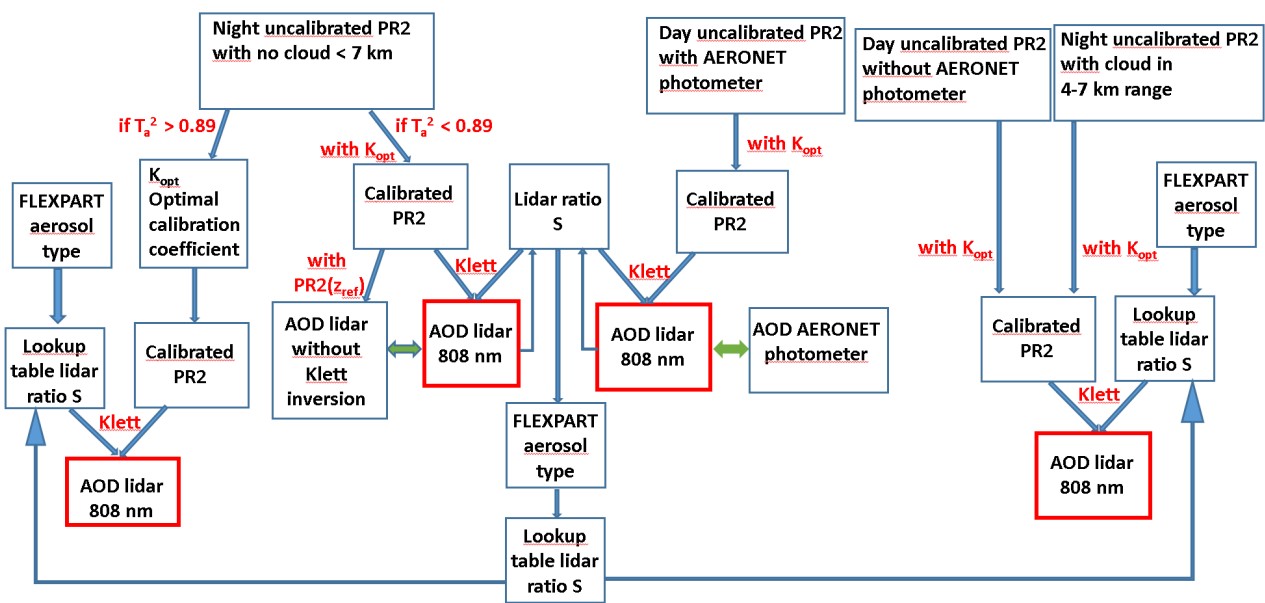

**Figure 4.** Flowchart of the lidar range-corrected signal (PR2) processing to derive the 800 nm aerosol optical depth (AOD) up to the reference altitude ($z_{ref}$). Four different calibration and AOD calculation are used according to the measurement conditions. Iteration between the AOD calculation and the lidar ratio value is only possible when the AOD can be compared to an external AOD reference (green two-sided arrows).

4. It is necessary because the cloud free lidar profile identified during the 18-month period cannot be always constrained by a sun-photometer AOD: nighttime observations, daytime observations without sun-photometer. Two different cases are identified in Fig. 4 for nighttime observations. First a direct AOD retrieval is possible when a layer with molecular signal can be found in the upper troposphere above 7.5 km and taking advantage of the good lidar calibration to measure the AOD from the lidar attenuated backscatter (see Appendix A). For the second case (cloudy conditions 4-7 km or very low AOD < 0.04 resulting in large error on $T_a^2$ deduced from the attenuated backscatter above 7 km ), we cannot rely on an independent estimate of $T_a^2$ to iterate on the proper lidar ratio. Assumption about the aerosol source must then be made, e.g. using FLEXPART simulations (see section 3) and the lidar ratio is taken from a lookup table for the five main Siberian aerosol sources. This lookup table is representative of Siberia because it is built using all the daytime and nighttime observations when independent $T_a^2$ are available.

Independent daytime $T_a^2$ at 808 nm can be obtained using the 870 nm AOD and the Angstrom coefficient (AE) measured by the AERONET sun-photometer located either on the Tomsk site (56.4°N, 85.0°E) or that of Tomsk22 (56.4°N, 84.1°E). Long-range transport of aerosol plumes are generally similar at both sites (Zhuravleva et al., 2017). There are 210 cases out of 539 lidar profiles with coincident lidar and sun-photometer observations.

To increase the number of lidar profiles constrained by an independent estimate of $T_a^2$, direct lidar measurement of the 808 nm AOD can be also obtained during the night if the lidar is well calibrated and if the reference altitude is above 7.5 km, i.e. with negligible contribution of particle backscatter (<10% of molecular backscatter). Indeed the value of the attenuated

backscatter ratio at altitude $z_r$ is then a direct measurement of the two-way transmittance $T_a^2(z_r)$ (Appendix A). The accuracy of the corresponding AOD is $\frac{1}{2}\frac{\Delta K}{K}$ = 4% when using the 8% accuracy on the calibration factor determined in section 2.2. The analysis is limited to AOD >0.04 to avoid large relative error on the retrieved AOD. There are 63 such cases, providing additional constraint for the lidar ratio retrieval.

## 3 Aerosol source attribution

Since there is no Raman channel on the CIMEL lidar, it is necessary to assess the likely variability of the aerosol sources to estimate the variability of the lidar ratio. A good knowledge of the aerosol sources linked to the lidar observations will be also beneficial to the analysis of the variability of the backscatter ratio and the AOD discussed in sections 5, 6. Backtrajectory analysis are widely used to identify the aerosol sources when the emissions area are well known. Our work is based on a similar approach but the improvement is to use the FLEXible PARTicle dispersion model (FLEXPART) version 9.3 to improve the likelihood of aerosol emission above the lidar site.

### 3.1 FLEXPART aerosol tracer simulation

FLEXPART is a Lagrangian model designed for computing the long-range transport, diffusion, dry and wet deposition, of air pollutants or aerosol particles backward or forward from point sources using a large number of particles (Stohl and Seibert, 1998; Stohl et al., 2002). Particle dispersion model calculations can be performed assuming two modes of transport in the atmosphere: passive transport without removal processes and transport of aerosol tracer, including removal by dry and wet deposition in the cloud and under the cloud (Stohl et al., 2012; Kristiansen et al., 2016). For each lidar profile, the latter was chosen using backward simulations of 10000 particles released in two altitude zones: (i) 500 m to $z_{aer}$ (ii) $z_{aer}$ to $z_{max}$, $z_{max}$ being the highest altitude with a scattering ratio R > 2 and $z_{aer}$ being the aerosol weighted altitude calculated with the aerosol backscatter vertical profile:

$$z_{aer} = \frac{\sum_{100m}^{z_r} \beta_a(z_i).z_i}{\sum_{100m}^{z_r} \beta_a(z_i)} \qquad (2)$$

For dry removal, particle density, aerodynamic diameter and standard deviation of a log-normal distribution were assumed to be 1400 kg m-3, 0.25 $\mu$m and 1.25, respectively following Stohl et al. (2013). Below-cloud scavenging is modeled using a wet scavenging coefficient defined as $\lambda = AI^B$, where A is the wet scavenging coefficient, I the precipitation rate in mm h-1, and B is the factor dependency. We set A=2.10$^{-5}$ s$^{-1}$, B=0.8. The in-cloud scavenging is simulated using a scavenging coefficient defined as $\lambda=(1.25I^{0.64})H^{-1}$, where H is the cloud thickness in m. The occurrence of clouds is calculated by FLEXPART using the relative humidity fields. The meteorological fields used for the simulations (including precipitation rates) are ERA Interim ECMWF field at T255 horizontal resolution ($\approx$ 80 km) and 61 model vertical levels.

A backward run of the model initialized from the receptor point (the lidar location) provides every 6 hours potential emission sensitivity (PES) fields in s with a vertical resolution of 1000 m and a horizontal resolution of 1.75$^o$x 1$^o$ (Seibert and Frank,

2004). These PES fields are generally recombined over a 9-day period either in the first vertical layer (0-1000 m) to obtain $PES_{surf}$ or over the first 5 vertical layers (0-5000 m) to obtain $PES_{0-5km}$. The first 12 hours before release are excluded to avoid a strong bias by the high PES due to recent local emissions which will mask high PES from remote sources. Examples of $PES_{0-5km}$ fields are shown in the section 5.1.

## 3.2 Distribution of aerosol sources

Several potential aerosol sources have already been identified for Siberia: (1) urban pollution (Dieudonné et al., 2017; Raut et al., 2017) , (2) flaring in the oil/gas industry (Stohl et al., 2013; Huang and Fu, 2016), (3) biomass burning (Warneke et al., 2009; Teakles et al., 2017) (4) dust from Central Asian deserts (Gomes and Gillette, 1993; Hofer et al., 2017), (5) organic aerosols emitted by taiga (Paris et al., 2009). The position of these source zones are coupled with the PES maps calculated by FLEXPART for the aerosol source attribution to a given lidar observation.

The role of urban pollution will be identified by the position of cities of more than 500,000 inhabitants in Russia, Mongolia and Kazakhstan without including neither emission inventory nor seasonal variation of the emissions. We are aware it is a crude assumption for a true aerosol modeling exercise but it a reasonable criteria to test the potential role of urban aerosol on the lidar data.

The biomass burning emission zones are derived from the Fire Radiative Power (FRP) daily maps provided by NASA Fire Information for Resource Management System (FIRMS) using MODIS (Giglio et al., 2003) and the Visible Infrared Imaging Radiometer Suite (VIIRS) (Schroeder et al., 2014). The FRP is estimated from both MODIS and VIIRS hot spots of the brightness temperature measurements. MCD14ML collection 6 standard quality products and VNP14IMGTDLNRT are used for, respectively, MODIS and VIIRS. The FIRMS data set then provides day (MODIS, VIIRS) and night (VIIRS) measurements with a spatial resolution of 1 km (MODIS) or 0.375 km (VIIRS). Only FRP values $> 0.3$ GW for MODIS and $> 0.1$ GW for VIIRS are used to identify biomass burning zones.

To identify continental regions covered by forests and deserts, we use the built-in United-States Geological Survey (USGS) 24 category land-use database in WRF (Weather Research and Forecasting) model. This global land cover database is derived from the Advanced Very High Resolution Radiometer (AVHRR) data with a resolution of 1 km spanning a 12-month period (April 1992-March 1993) (Sertel et al., 2010). The role of dust plumes can be overestimated when using only the land-use map, so it is only considered if neither urban pollution nor biomass burning have been identified.

Russia and Nigeria are the two biggest contributors to gas flaring used at oil/gas production and processing sites. The location of flaring sources is based on the anthropogenic emissions ECLIPSEv4 database (Evaluating the Climate and Air Quality Impacts of Short-Lived pollutants) described in Klimont et al. (2017). This inventory includes in particular the gridded methane emissions from gas flaring in the Russian Arctic at a 0.5° x 0.5° deg horizontal resolution. A threshold of 50 moles/km$^2$/hour has been applied to the methane emissions to select areas that could potentially be defined as flaring sources. Owing to the strong variability of flaring emissions, the role of flaring may be overestimated, so, as we do for the dust emission, it is only considered if anthropogenic and biomass burning sources are not identified.

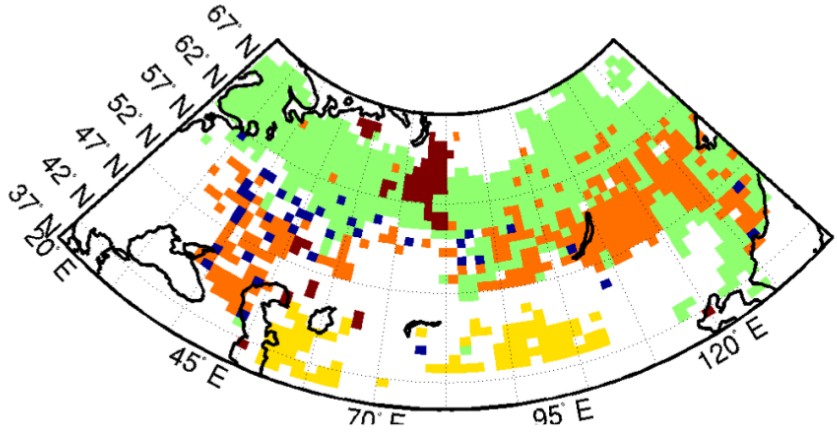

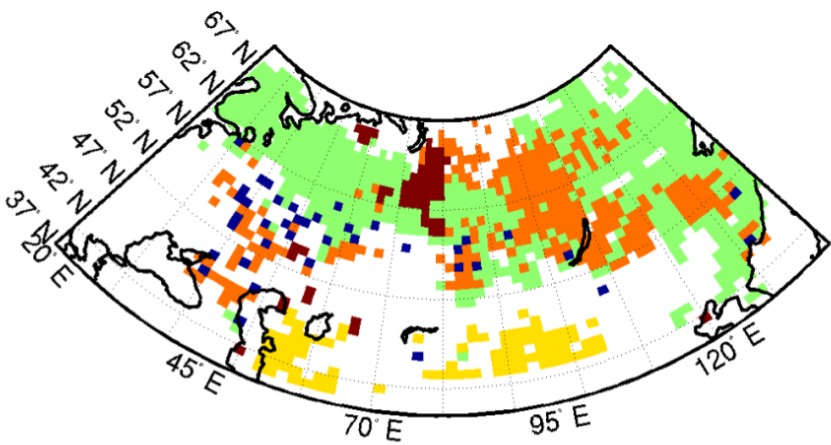

**Figure 5.** Map of the 2015 (top) and 2016 (bottom) aerosol sources coupled with FLEXPART PES gridded map: grid cells with large cities (blue square), Central Asia desert (yellow), biomass burning (light brown), gas flaring (dark brown), taiga forest (green).

The map of the main aerosol emission sources are shown in Fig. 5 for, respectively, 2015 (top) and 2016 (bottom). The lidar measurement site corresponds to the blue square at 56°N, 85°E. In 2016 forest fires were very numerous in Central Siberia while they are much further east of Lake Baikal in 2015. When $PES_{surf} > 1500$ s for at least one grid cell with a large city or flaring emissions, the type of aerosol is classified as, respectively, urban aerosol, or flaring aerosol. When $PES_{0-5km} > 1500$ s for at least one grid cell with fires or desert soils, the type of aerosol is classified as, respectively, biomass burning aerosol, or dust aerosol. $PES_{0-5km}$ is chosen for dust and biomass burning plume which can be quickly uplifted in the free troposphere up

to 5 km. If none of the above conditions are fulfilled, the remaining significant source is the contribution of oxygenated aerosol emission from the very large area covered by the Taiga forest (Zhang et al., 2007).

## 4 Lidar aerosol optical depth: Results

### 4.1 Nighttime direct AOD measurements

The probability density function (PDF) of the nighttime lidar AOD using the direct retrieval method described in section 2.3 is compared with the PDF of the AOD measured during the day by the sun-photometer (not including one third of the sun-photometer AOD < 0.04 since AOD < 0.04 are not considered in the direct nighttime AOD retrieval). The comparison shows that our nighttime retrieval using the backscatter ratio at $z_r$ gives a realistic distribution of the AOD with similar median and 90th percentile of the AOD (Fig. 6a,b). A direct comparison between nighttime lidar AOD and photometer AOD is not

possible in Tomsk because lunar photometry is not available. The alternative solution is to use sun-photometer AOD with a time difference < 6 hours with the lidar observations and to include the observed daily variability of the sun-photometer AOD. The correlation plot is also shown in Fig. 6c showing no clear bias and a satisfactory agreement considering the daily variability of the sun-photometer AOD (error bar in Fig. 6c) .

### 4.2 Integrated lidar ratio retrieval

According to Fig. 4, backward inversion of the calibrated lidar attenuated backscatter can be done iteratively using different lidar ratios when the optical thickness calculated with the extinction profile is compared with the independently obtained AOD. The final solution is always obtained after 6 iterations. Starting with the largest expected lidar ratio allows a fast convergence towards the true value (e.g. see Young (1995)). Thirteen $S_{808} < 45$ sr could be retrieved with this method out of the 15 FLEXPART dust cases even though iteration starts with 60 sr. A set of 273 lidar ratio constrained by daytime observations

with sun-photometer or by nighttime measurements where $T_a^2(z_r)$ is then available to built the lidar ratio lookup table (Tab. 1) for each of the five aerosol types determined with the FLEXPART analysis described in Section 3 and for three seasons: cold season (15/10 to 15/3), spring (15/3 to 30/6) and warm season (30/6 to 15/10) The standard deviation of the lidar ratio for each class is a good proxy for the error on the 273 $S_{808}$ values retrieved with this method. Since the 10 sr error remains significant, it is important to discuss the lidar variability obtained in Table 1. First as expected (Omar et al., 2009; Burton et al., 2012), the

lowest values (40 sr) are indeed obtained for the desert aerosol class, while the highest values (>60 sr) are characteristic of pollution aerosols (flaring and urban pollution in winter). Using the spectral variability of the lidar ratio proposed by Cattrall et al. (2005) to calculate the equivalent S values at 532 nm, $S_{532}$ is 50 sr for the lower limit of our lidar ratio and 80 sr for the lidar ratio of pollution aerosol. This is consistent with the Burton et al. (2012) analysis, but the lower limit is higher than the average lidar ratio obtained by Hofer et al. (2017) (35 sr) in the deserts of Tajikistan. Aerosol growth and mixing during long

range transport is likely responsible for higher values of S in Tomsk (Nicolae et al., 2013; Ancellet et al., 2016)

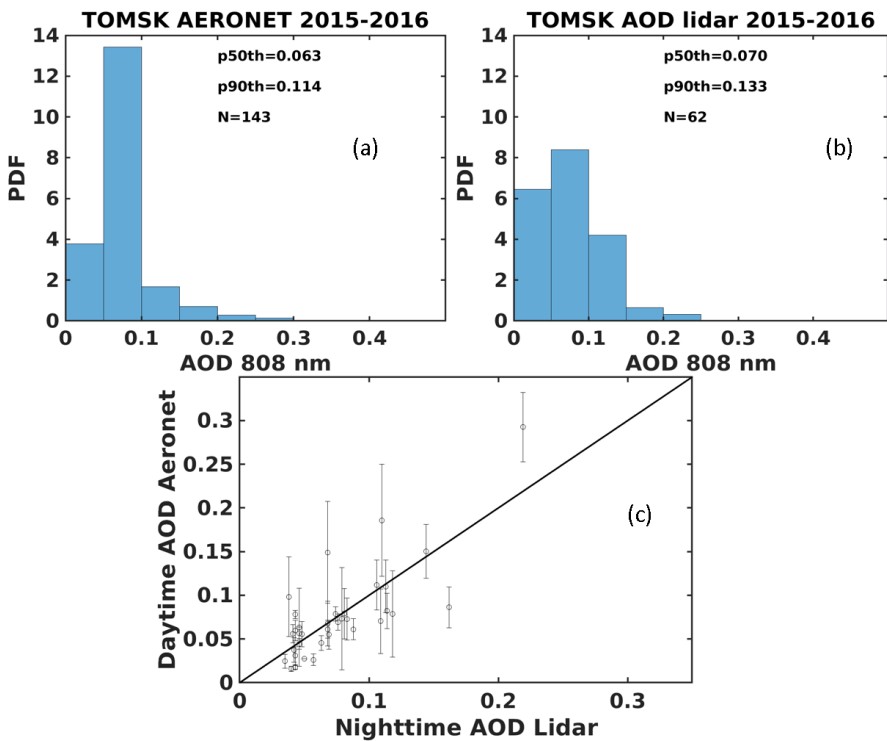

**Figure 6.** (a) PDF of daytime AOD from sun-photometer for lidar measurement days and (b) of nighttime AOD calculated with the lidar attenuated backscatter ratio measurement at the reference altitude $z_r$. N is the number of observations, while p50th and p90th are respectively the median and $90^{th}$ percentile of the AOD distribution. (c) Correlation plot of nighttime lidar AOD versus sun-photometer daytime AOD when the time difference is less than 6 hours between the two measurements (35 cases out of 63 nighttime lidar AOD). The error bar is the daily variability of the sun-photometer AOD.

For the remaining 267 lidar profiles where the lidar ratio cannot be constrained by the sun-photometer or a good calibrated lidar measurement above 7.5 km, the FLEXPART analysis and the lidar ratio lookup table (Table 1) are used to retrieve the backscatter ratio and the extinction profile (see right and left side cases in Fig. 4) . The relative error on the AOD calculated with the extinction profile is then mainly related to the relative error on the lidar ratio derived from the lookup table, i. e. of the order of 25%.

### 4.3 Lidar AOD seasonal variability

The whole time series of the median of the backscatter ratio $R_{808}$ between 0-2.5 km and 2.5-5 km are shown in Fig. 7. As expected the mean backscatter ratio >3 are seen mainly in the lowermost troposphere below 2.5 km (22% of the 540 profiles), while only 5% are observed for the altitude range 2.5-5 km. Elevated backscatter ratio (>3) are observed from February to

**Table 1.** Lidar Ratio at 808 nm in sr for the 5 FLEXPART derived aerosol types and 3 seasons (cold,spring and warm) when using independent AOD measurements.

| Season | 15/10 to 14/3 | 15/3 to 30/6 | 1/7 to 14/10 |
| | Cold | Spring | Warm |
| --- | --- | --- | --- |
| Urban | 61±10 | 51±15 | 46±11 |
| Flaring | 70±10 | 61±12 | 52±15 |
| Biomass Burning | 54±14 | 57±14 | 50±15 |
| Dust | 42±10 | 46±9 | 36±9 |
| Taiga | 52±15 | 50±16 | 56±14 |

September below 2.5 km and from April to September in the free troposphere. The latter is more or less in phase with the start/end date of dust storm and forest fires periods in Eurasia.

The time series of the AOD calculated from the extinction vertical profiles is then compared to the AOD from the sun-photometer (Fig. 8). The agreement is generally good between the two time series of AOD and elevated AOD ($> 0.2$) are clearly visible at about the same periods. More short term variability is obtained for the sun-photometer AOD since all 10-min cloud free observations are shown in Fig. 8. The elevated AOD are not only observed in summer (June to September), which indicates that biomass burning episodes are not solely responsible for the strong AOD. A strong difference between $AOD_{550}$ for warm (AOD=0.3) and cold season (AOD=0.08) has been also reported by Chubarova et al. (2016) for the city of Moscow. The corresponding time evolution of the aerosol weighted altitude calculated with Eq. 2 shows an average altitude of 1.5 km, meaning that the major contribution of the extinction profile to AOD is within the altitude range 0-2.5 km defined hereafter as the Planetary Boundary Layer (PBL). For periods with elevated AOD, e.g. A, B, C in Fig. 8, $z_{aer}$=2, 3.5, 1 km, respectively. So $z_{aer}$>2 km is not only related to an aerosol extinction profile with low AOD.

## 5 Case studies: comparison lidar, sun-photometer, satellite observations

In this section, we focus on the time periods with elevated AOD observed by the AERONET network above Tomsk in order to (1) compare the results of our AOD analysis with AERONET values during 48 h around the selected lidar profiles and with satellite data (MODIS or CALIOP) (2) identify the likely aerosol sources derived from the FLEXPART analysis with satellite observations (MODIS, IASI, CALIOP) in the source areas . Looking at Fig. 8, there are 5 time periods with sun-photometer AOD $> 0.2$: mid-may 2015, end of may 2015, April 2016, mid-June 2016 and end of September 2016. We do not have enough lidar data for mi-June 2016. The end of September 2016 and mid-June 2015 cases both correspond to forest fire events, while end of may 2015 and April-2016 correspond to urban, flaring and dust emissions according to our FLEXPART analysis. Therefore the three time periods corresponding to periods A, B, C of Fig. 8 are analyzed in this section. The section

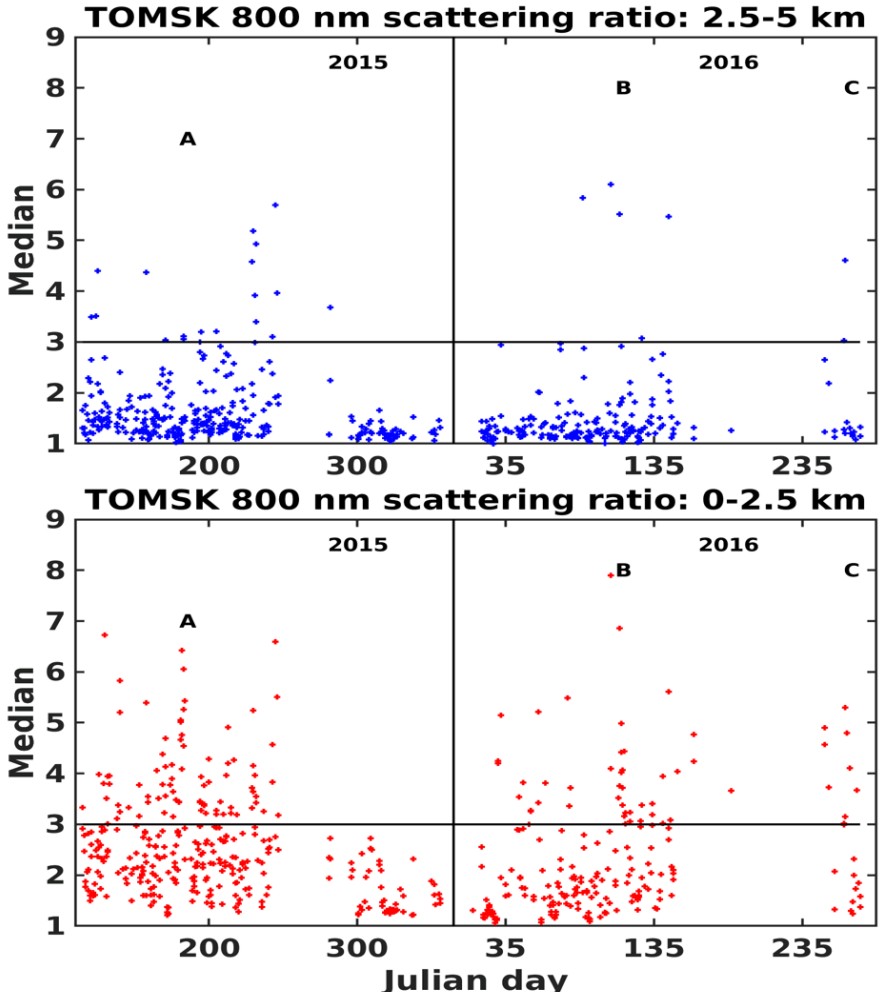

**Figure 7.** Time evolution from April 2015 to September 2016 of the median of the 808 nm Tomsk lidar backscatter ratio for two altitude ranges: 0-2.5 km (bottom) and 2.5-5 km (top). A, B, and C are the cases analyzed in section 5.

5.1 presents the daily variability of the lidar backscatter profiles and sun-photometer AOD, while the section 5.2 presents the analysis of satellite observations.

### 5.1 Lidar data daily variability and comparison with sun-photometer AOD

From 14 to 15 April 2016 (case B in Fig. 8), $AOD_{808}$ varies between 0.05 and 0.3 for both the lidar and for the sun-photometer
5 (Fig. 9a). The vertical profiles of the backscatter ratio (Fig. 10) show a tripling of the aerosol content in the PBL in 12 hours which is consistent with the daily variability of the sun-photometer AOD. The comparison of the two nighttime lidar profiles shows also a similar increase of the aerosol content between 2.5 and 5 km altitude, which suggests that long range transport of

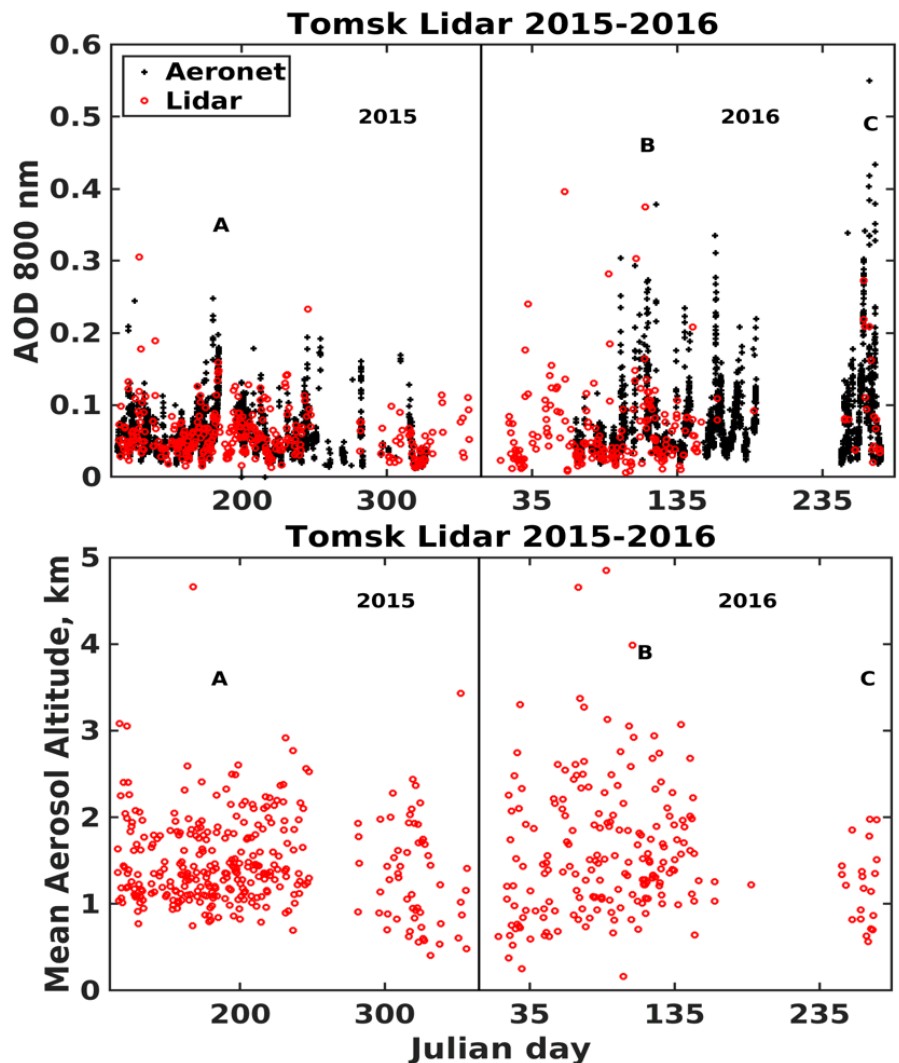

**Figure 8.** Time evolution from April 2015 to September 2016 of the 808 nm AOD for lidar (red) and sun-photometer (black) observations (upper panel) and corresponding aerosol altitudes $z_{aer}$ given by Eq. 2 (bottom panel). A, B, and C are the cases analyzed in section 5

aerosol layer took place above the PBL. $S_{808}$ decreases from 60-70 sr to 42 sr along with the AOD increase, showing that the increase of aerosol concentrations is the driving factor for the tripling of the AOD. The FLEXPART simulation (Fig. 10) shows strong PES values (>1500 s) northwest of Tomsk over the Ob industrial valley between Tomsk (56°N, 85°E) and Surgut (62°N, 73°E) for aerosols detected below 2.5 km. The strong PES values are much more scattered for the upper layer above 2.5 km

5    with aerosol sources both from the lower Ob valley and from a large part of Kazakhstan. Indeed according to our classification of the type of aerosol, measurements below 2.5 km have been classified as flaring on 14 April, urban on 15 April 3 UT and

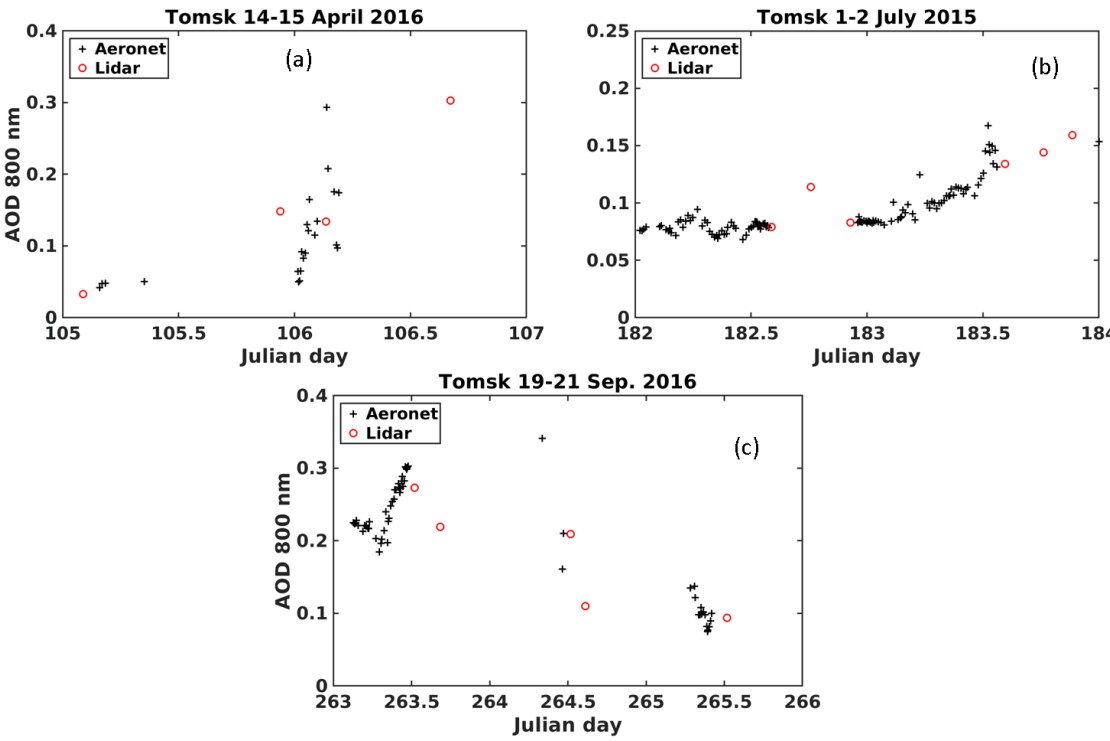

**Figure 9.** Diurnal evolution of the lidar AOD and the sun-photometer AOD at 808 nm for A (b), B (a) and C (c) cases shown in Fig.8.

dust on 15 April 16 UT. Measurements above 2.5 km were classified as dust emissions. The decrease of $S_{808}$ is also consistent with a decreasing fraction of pollution aerosol when dust is advected from Kazakhstan (Burton et al., 2012; Hofer et al., 2017).

From 30/6 to 2/7/2015 (case A in Fig. 8), the measured $AOD_{808}$ also gradually increase from 0.08 to 0.16 for both the lidar and the sun-photometer (Fig. 9b). The vertical profiles of the backscatter ratio (Fig. 11) show as in the previous case high values (5-7) in the PBL but lower values ($\approx$ 4) above 2.5 km. $S_{808}$ increases from 35 sr to 47 sr, implying that the AOD increase is due both to a change of the aerosol type (40%) and an increase of the aerosol load (60%). PES maps indicate an origin still associated with the lower Ob valley for measurements in the PBL (Fig. 11), while high PES values are observed between Tomsk and lake Baikal for the layer observed in the free troposphere. The Baikal region was impacted by forest fires end of June 2015 (see Section 5.2), so our classification indeed indicates biomass burning aerosol for the layer above 2.5 km and a mixture of aerosol produced by flaring (30/6 and 1/7) and by biomass burning (2/7) in the PBL. Although the $S_{808}$ increase is consistent with the advection of biomass burning aerosol, $S_{808}$ is surprisingly low (35 sr) for the plume advected at the beginning of the period from the flaring region. One explanation is the strong daily variability of flaring emissions which cannot be taken into account for our flaring type attribution only based on advection from the flaring region. $S_{808}$ of 47 sr is also in the lower range of expected value for biomass burning, in accordance with the dual air mass origin for the upper layer in Fig. 11, which implies some aerosol mixing.

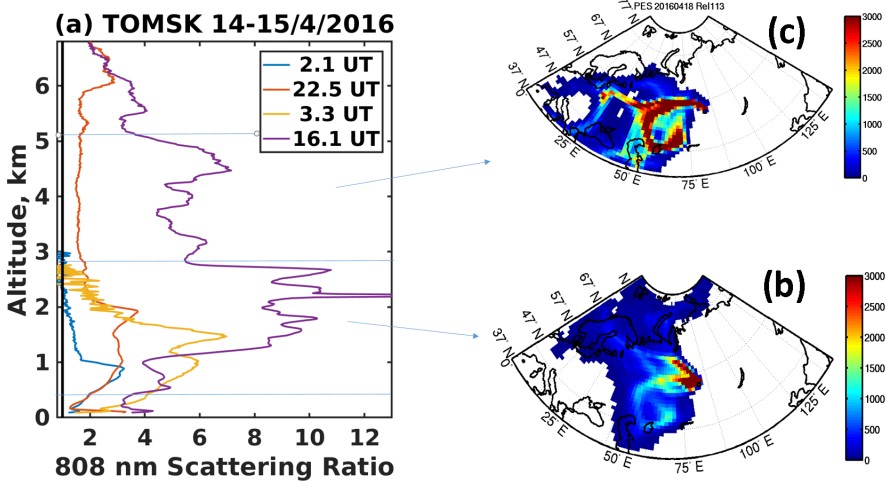

**Figure 10.** Vertical profiles of the scattering ratio on 14 April 2016 at 2.1 UT and 22.5 UT, and on 15 April 2016 at 3.3 UT and 16.1 UT (a) and map of the PES distribution for FLEXPART backward simulation initialized in the PBL (b) and above the PBL (c)

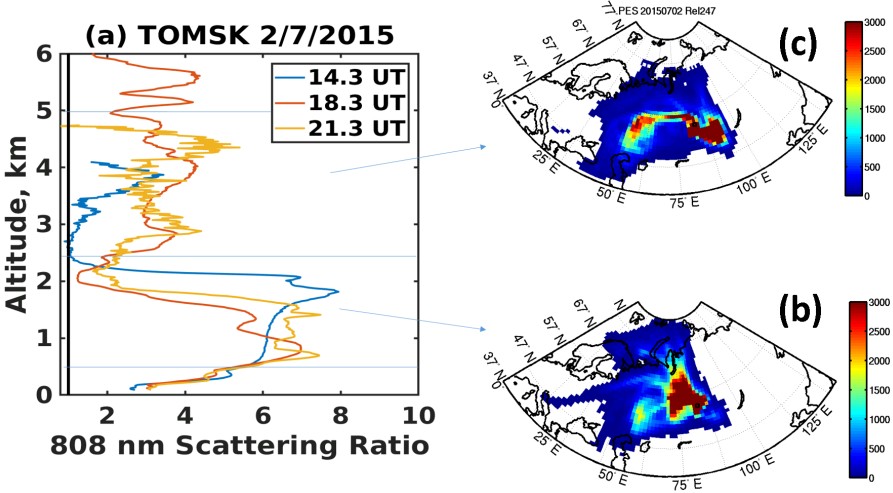

**Figure 11.** as Fig. 10 on 2/7/2015 at 14.3 UT, 18.3 UT and 21.3 UT

From 19 to 21 September 2016 (case C in Fig. 8), $AOD_{808}$ values were decreasing from 0.4 to 0.1 according to the sun-photometer and the lidar (Fig. 9c). The vertical profiles of the backscatter ratio (Fig. 12) actually show a very strong decrease in the PBL (20 to 5), with the high values being confined in the 0-800 m altitude range. The aerosol content above 1 km is lower with $R_{808}$ ranges between 3 and 5. $S_{808}$ always remains about 60 sr except at the end of the 3-day period where it drops to 40 sr for AOD equal to 0.1. The PES distributions are different from the 2 previous cases with a large horizontal extension of the area with strong PES values for the PBL (Fig. 12). This area includes a 500 km circle around Tomsk and two branches

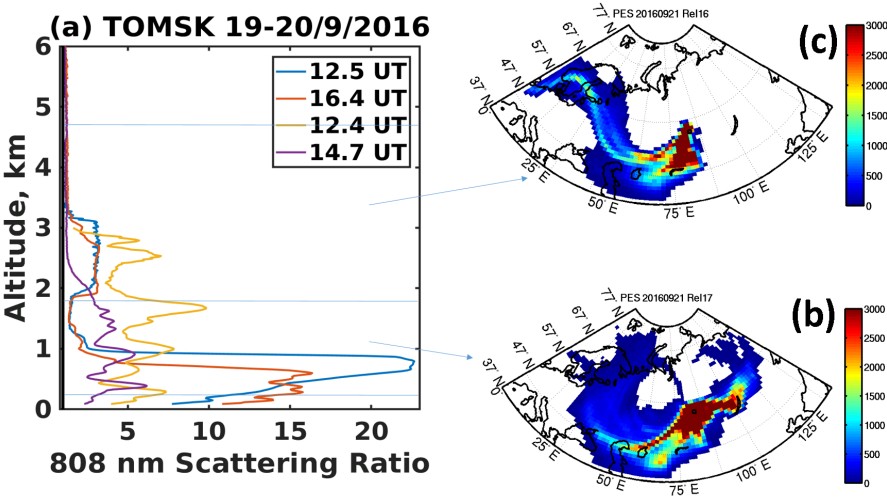

**Figure 12.** as Fig. 10 on 19/9/2016 at 12.5 UT and 16.4 UT and on 20/9/2016 at 12.4 UT and 14.7 UT

extending on the one hand to Lake Baikal and on the other hand to Kazakhstan. On the contrary, aerosol sources are now confined, for the free troposphere, to a south-west sector above Novosibirsk and Kazakhstan (Fig. 12). For the entire period 19 to 21 September, aerosols were classified as biomass burning aerosols due to the presence of forest fires over a large area to the east and north of Tomsk.The 60 sr high values of $S_{808}$ are consistent with the transport of the biomass burning aerosol
from Eastern Siberia (Burton et al., 2012), while even the $S_{808}$ drop to 40 sr is explained by the mixing with air coming from Kazakhstan.

## 5.2 Satellite observations

### 5.2.1 Description of data products

Available satellite observations for these three periods were selected to identify the aerosol source regions. The horizontal
distribution of strong AOD is documented by the 550 nm MODIS AOD maps averaged over 5 days. AOD maps are made using the Level-3 MODIS Atmosphere Daily Global Product which contains roughly 600 statistical datasets sorted into 1 by 1 degree cells on an equal-angle grid that spans a 24-hour interval (Platnick et al., 2015; Levy et al., 2013). The role of biomass burning or fuel combustion can be described with satellite tropospheric CO column measured e.g. by the Infrared Atmospheric Sounding Interferometer (IASI) instrument on Metop A and B. Because a large fraction of atmospheric CO is also related to
the oxidation of hydrocarbons including methane, flaring will be a source of CO. The IASI CO data used in this paper have been processed at LATMOS using a retrieval code, FORLI (Fast Optimal Retrievals on Layers for IASI), developed at ULB (Université Libre de Bruxelles) by Hurtmans et al. (2012). Validation for Siberia and Arctic region is described in Pommier et al. (2010). The vertical distribution of aerosol layers is inferred from CALIOP overpasses. In this work 532 nm backscatter and depolarization ratios are calculated using the CALIOP level-1 (L1) version 4.10 attenuated backscatter coefficients because

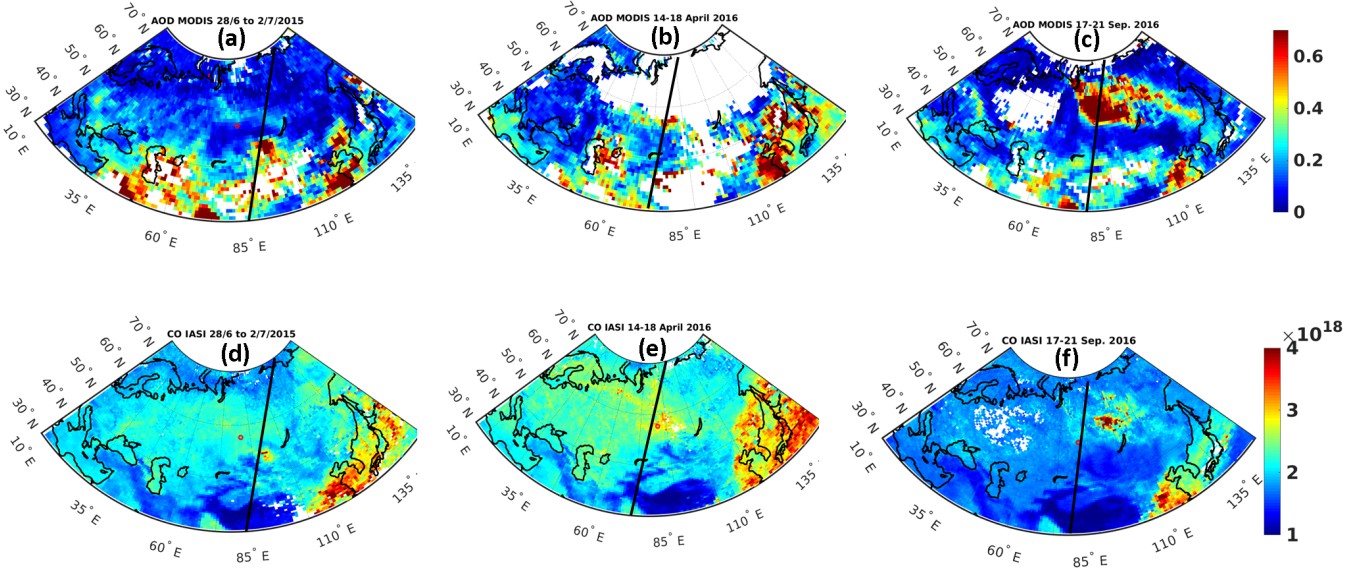

**Figure 13.** Five day average of AOD at 532 nm from 1°x 1° MODIS observations (a,b,c) and of CO in molecules.cm$^{-2}$ from IASI observations (d,e,f) for July 2015 (a,d) April 2016 (b,e) and September 2016 (c,f). The red circle is Tomsk and the black thick lines are the CALIOP overpasses shown in Fig. 14 to 16.

they correspond to a better calibration of the lidar data (Vaughan et al., 2012; Winker et al., 2009). They are averaged using a 10 km horizontal resolution and a 60 m vertical resolution. Before making horizontal or vertical averaging, the initial 333 m horizontal resolution (1 km above the altitude 8.2 km) are filtered to remove the cloud layer contribution. This cloud mask makes use of the Version 3 level-2 (L2) cloud layer data products (Vaughan et al., 2009) and measurements of the IR imager

on the CALIPSO platform. Our scheme for distinguishing cloud and aerosol is described in Ancellet et al. (2014). To calculate the extinction profile and the optical depth, we use the lidar ratio $S_{532}$ from the CALIOP Version 3 L2 aerosol layer data products (Omar et al., 2009), unless we can calculate the aerosol layer transmittance to constrain $S_{532}$. To reduce the error when using high horizontal resolution CALIOP profiles, the attenuated backscatter is averaged over 80 km to compute the layer transmittance whenever it is possible. The aerosol depolarization ratio $\delta_{532}$ is also calculated using the perpendicular- to

the parallel plus perpendicular polarized aerosol backscatter coefficient (see Appendix B). Whenever it is possible, the use of nighttime overpasses are preferred to improve the signal-to-noise ratio (SNR).

### 5.2.2 Results

From 14 to 18 April 2016, the AOD MODIS and CO IASI maps (Fig. 13b,e) show maxima around the town of Tomsk and more generally in the lower Ob valley (only for IASI insofar as the cloud cover and snow cover do not allow MODIS to be

used above 58°N). No forest fires were detected during this time period and a predominant role of flaring emissions seems a likely hypothesis for the aerosol layers observed at Tomsk. A CALIPSO overpass with low cloud cover between 50°N and

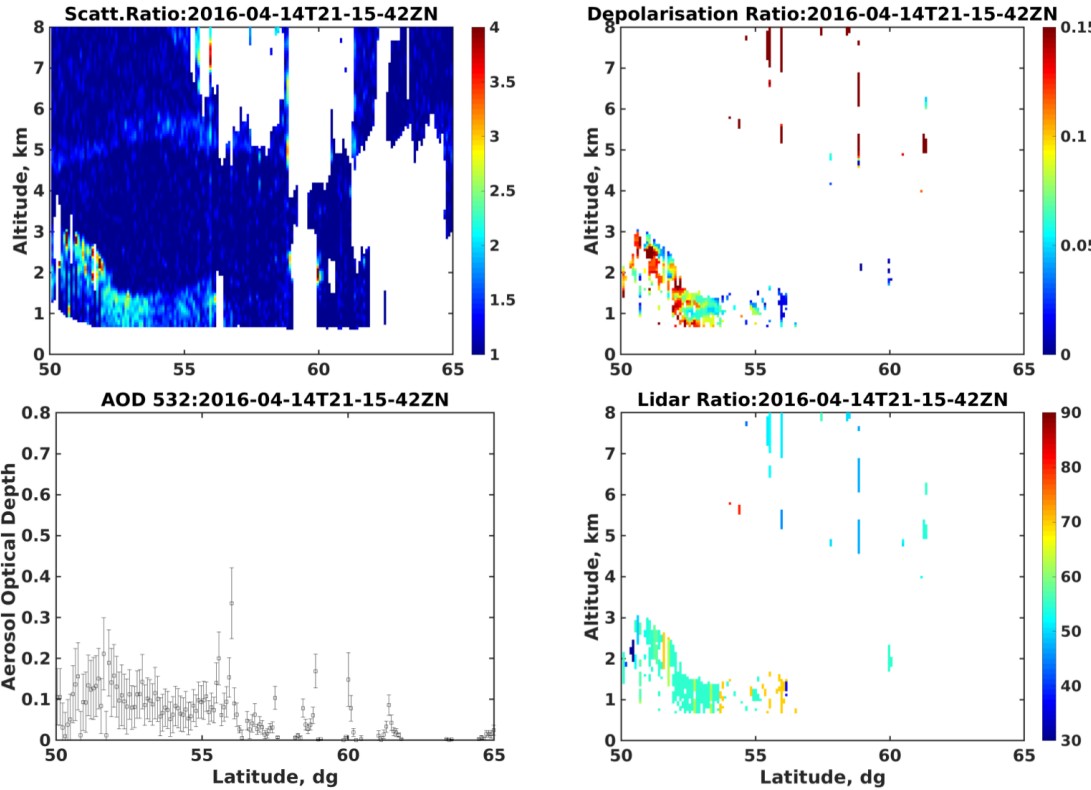

**Figure 14.** Latitudinal cross-section of CALIOP 532 nm scattering ratio $R_{532}$ (top left), aerosol depolarization ratio $\delta_{532}$ (top right), aerosol optical depth $AOD_{532}$ (bottom left) and lidar ratio $S_{532}$ (bottom right) for 14/4/2016.

60°N is available on 14/4/2016 (thick black line in Fig. 13). The $AOD_{532}$ observed by CALIOP (Fig. 14) in the range 0.05-0.15 and the associated backscatter ratio ($\approx 2$) are lower than the highest values observed by the Tomsk lidar ($AOD_{808} \approx 0.3$, e.g. corresponding to $AOD_{532} \approx 0.4$ using the sun-photometer AE=0.87), but it is consistent with the range 0.07-0.4 of $AOD_{532}$ when using the $AOD_{808}$ observed by the TOMSK lidar. The CALIOP AOD is also lower than the 5-day average MODIS $AOD_{550}$ ($\approx 0.5$) near Tomsk (Fig. 13b), because the CALIOP track was on the edge of the MODIS AOD maxima. The CALIOP observations however provides the vertical (0-2 km) and latitudinal (52°N to 57°N ) extent of the aerosol layer due to flaring/urban emissions (Fig. 14), similar to the Tomsk lidar observations. At latitude below 52°N, an aerosol layer is identified as dust by CALIOP with $AOD_{532} \approx 0.2$, an upper boundary up to 3 km and depolarization ratio >12%. The lidar ratio attributed by CALIOP is 55 sr, being consistent with dust emission from Kazakhstan being responsible for the increasing AOD and advection of the aerosol layer observed in Tomsk above 2.5 km on 15 April 2016.

From 28/6 to 2/7/2015, the MODIS AOD and IASI CO maps (Fig. 13a,d) indicate two maxima with both elevated $AOD_{550}$ and CO values: a forest fire zone of $3.10^5$ km$^2$ at 51°N, 97°E ($AOD_{550} > 0.7$, i.e. $AOD_{808} \approx 0.3$ with AE=2), the flaring zone on the lower Ob valley between 56°N and 65°N ($AOD_{550} \approx 0.3$, i.e. $AOD_{808} \approx 0.13$ with AE=2). This is in rather good agreement

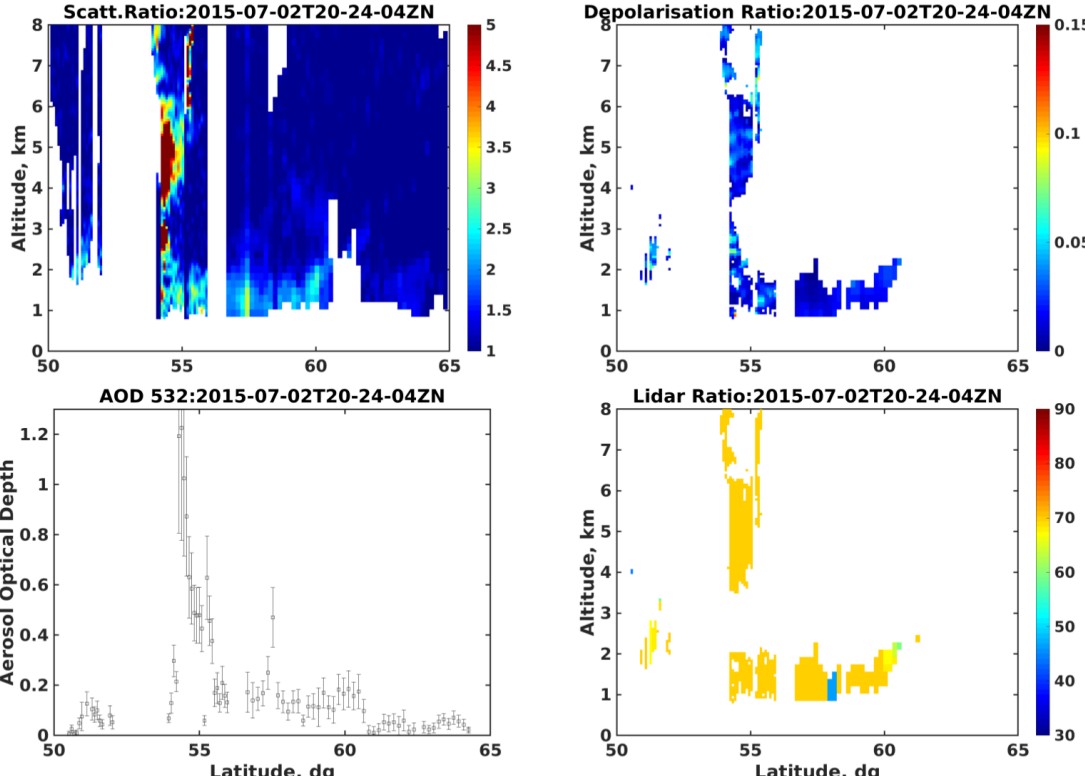

**Figure 15.** Same as Fig. 14 for 2/7/2015.

with our analysis of aerosol sources which indicates a mixture of fire and flaring emissions for aerosol layers observed below 2.5 km at Tomsk and a role of fires in the free troposphere above 2.5 km. There is only one CALIPSO overpass on 2/7/2015 (thick black line in Fig. 13) across the fire plume west of Lake Baikal. Elevated $AOD_{532} > 0.5$ are indeed observed by CALIOP at 54°N,97°E in smoke layers with very low depolarization ratio (<5%) and backscatter ratio >5 up to an altitude of 6 km (Fig.

5 15). The corresponding AOD at 808 nm of the order of 0.18-0.45, when using the 1.9 sun-photometer AE over Tomsk on 2 July 2015, shows that the Tomsk lidar AOD is two times lower after being transported from lake Baikal and mixed with background aerosol. It also explains the 47 sr moderate $S_{808}$ for a biomass burning event as discussed in section5.1. The satellite data analysis for July 2015 is therefore consistent with the results of the Tomsk lidar data processing both for the AOD range, and for the aerosol type assumption, and related lidar ratio.

10 From 17 to 21 September 2016, the AOD MODIS and CO IASI maps show a very large area impacted by the numerous forest fires (see `https://www.fire.uni-freiburg.de/GFMCnew/2016/09/28/20160928_ru.html`) that took place in Siberia in September 2016. MODIS $AOD_{550} > 0.7$ and CO columns $> 3\ 10^{18}$ mol.cm$^2$ are observed over an area of 1000 km x 1000 km at 57°N-67°N, 95°E-115°E Fig. 13c,d). Tomsk lies just at the edge of this wide plume. The influence of biomass burning aerosol found in our analysis of Tomsk lidar observations is linked to this event. The CALIPSO track

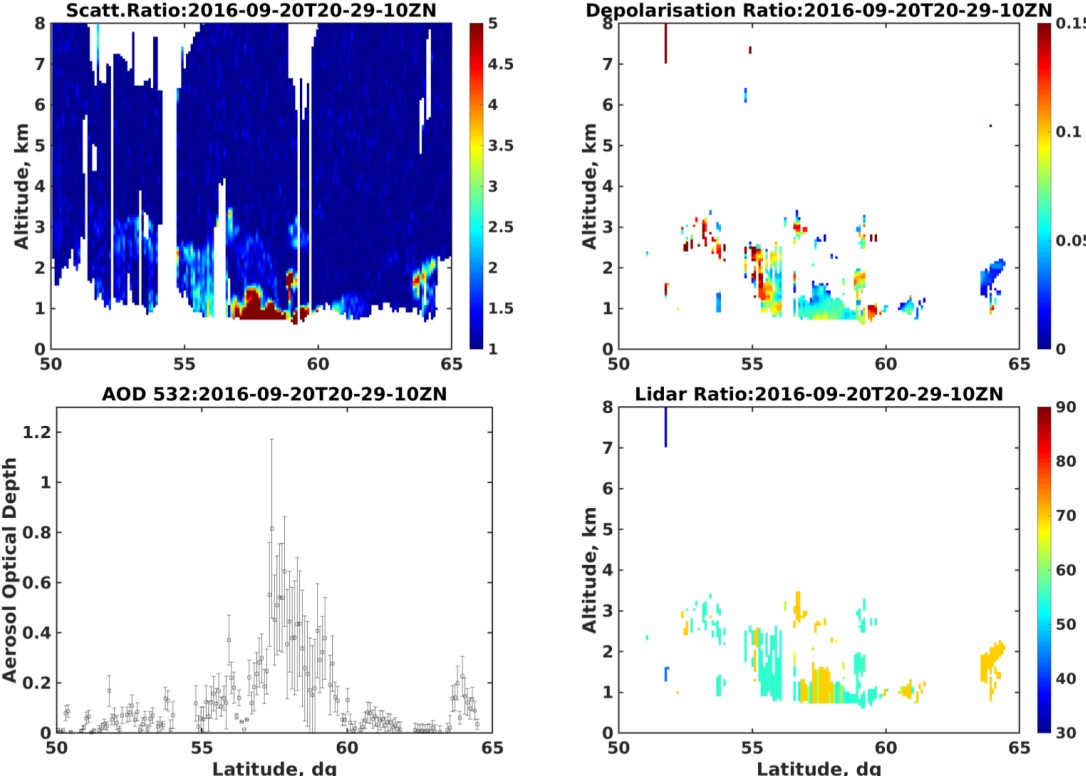

**Figure 16.** Same as Fig. 14 for 20/9/2016.

passing over Tomsk and over the fire plume between 56°N and 70°N (black line in Fig. 13c,d) shows $AOD_{532}$ in the range 0.4-0.8 between 56°N and 60°N (Fig. 16). The CALIOP AOD increase corresponds to the MODIS AOD anomaly, although it is 30% lower than the average MODIS AOD maxima (Fig. 13c). The MODIS AOD is consistent with $AOD_{808} > 0.4$ observed in Tomsk, i.e. a corresponding $AOD_{550} = 0.75$ using AE=1.5 as measured by the Tomsk sun-photometer during this time period.

5   The CALIOP depolarization ratio (7%) is higher than for the July 2015 fire event indicating soil aerosol vertical transport simultaneously with the production of biomass burning aerosol for these late summer fires (Nisantzi et al., 2014). Similarly the CALIOP lidar ratio ($\lesssim$70 sr) and the sun-photometer AE (1.5) are lower than the values obtained for the July 2015 fires ($S_{532} \approx 75$ sr, AE=1.9) even if these values remain characteristic of a combustion aerosol. The corresponding $S_{808} \lesssim 53$ sr for the 20/9 CALIOP cross-section is lower than $S_{808} = 60$ sr measured for the biomass burning plume in Tomsk, but the difference

10   is well within the 10 sr expected uncertainty for the Tomsk lidar $S_{808}$ and the known uncertainties for the CALIOP lidar ratio assessment (Omar et al., 2009). It is also interesting to see that the vertical extent of the fire plume observed by CALIPSO remains fairly low (<1.5 km) between 55°N and 60°N, i.e. an aerosol plume thickness similar to the Tomsk lidar measurement.

The overall conclusion of this section 5 is that (1) our approach to attribute an aerosol type to Tomsk lidar observations is validated by a more in-depth study of aerosol sources based on available satellite observations, (2) the AOD and lidar

ratio calculated for the Tomsk lidar observations are comparable to the sun-photometer daily AOD variability and satellite AODs in the aerosol source regions identified by the FLEXPART analysis. This will allow the statistical analysis of the lidar measurements according to aerosol types for the 18-month database.

## 6 Contribution of aerosol sources to aerosol optical depth distribution

In this section, all observations from April 2015 to October 2016 will be analyzed taking into account the type of aerosol source attributed to each aerosol layer in Section 3. The PDFs of AOD at 808 nm have been calculated for the different aerosol types determined with the FLEXPART analysis. To distinguish the AOD distribution for PBL only and PBL plus free tropospheric (FT) aerosol, the PDFs are shown for $z_{aer} <= 1.75$ km and $z_{aer} > 1.75$ km (Fig. 17). The results show that the distribution of AOD when including all aerosol types, has a median value of about 0.05 and a very rapid decrease in the number of observations when AOD$> 0.1$ ($90^{th}$ percentile of about 0.11). If the AOD distributions for the organic aerosol class emitted by vegetation (forest /grassland) and for flaring emissions are not significantly different from the AOD distribution for all types, those for the other classes (urban pollution, biomass burning, dust) have a dominant AOD mode closer to 0.1. The highest $90^{th}$ percentile (AOD$\geq$0.18) are for forest fire and dust emissions although the number of events is statistically lower for these aerosol types than for other emission sources.

The proportions of aerosol types calculated with the number of observations are indeed 41%, 28%, 16%, 10% and 5% for forest/grasslands emissions, urban pollution, flaring, biomass burning and dust respectively. The dust contribution is very weak as transport pathways and orography reduce significantly the northward transport of Central Asian dust plumes. If we consider only AOD$>$0.1, these relative proportions become very different: 25%, 25%, 10%, 27% and 13% for forest/grassland, urban pollution, flaring, vegetation fires and dust respectively. The dust emission contribution to large AOD values becomes now as large as the flaring emission contribution, and the biomass burning contribution becomes equivalent to urban or forest emissions.

Looking at the differences between PDFs for PBL only (blue) and PBL plus FT (red), the forest/grassland, forest fire and flaring emissions correspond to 60%-70% of the AOD measured in PBL while the proportion reaches 76% for urban emissions and drops to 35% for dust. This is consistent with urban aerosol emissions associated with the Tomsk/Novosibirsk/Kemerovo triangle being confined below 2.5 km while dust plumes associated with long-range transport mix little with the boundary layer. It should also be noted that although forest fire plumes are often associated with long-range transport, their incorporation into the PBL remains effective (70% of observed cases). Even when AOD is limited to values $> 0.1$, the proportion of biomass burning aerosol incorporated below 2.5 km remains high (66%), while that of urban aerosol decreases significantly from 76% to 53%.

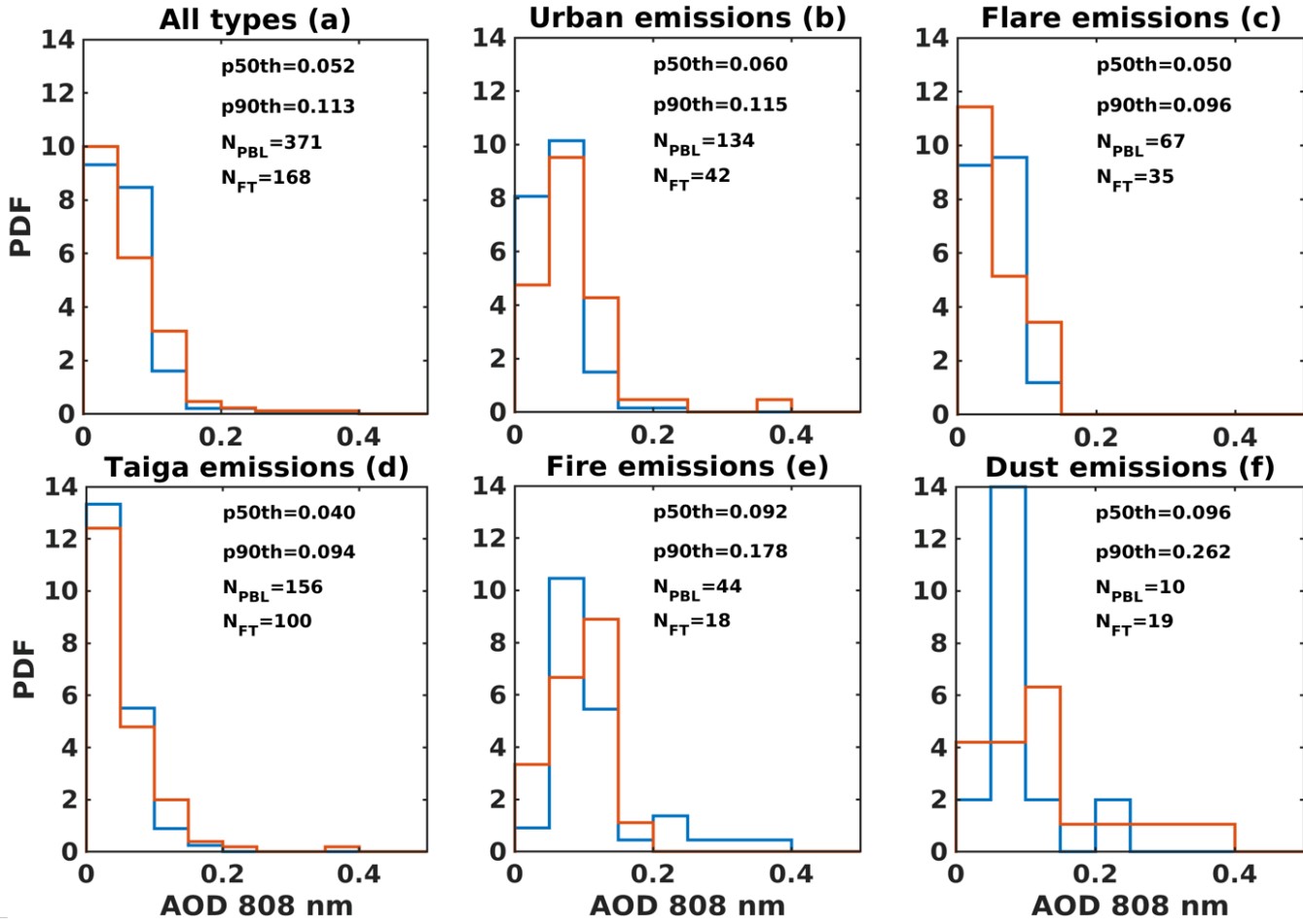

**Figure 17.** PDF of the 808 nm AOD lidar according to the different of aerosol types: all (a), urban (b) flaring (c), natural emissions from Siberian forest and grasslands (d) biomass burning (e) dust (f). Blue PDF is for aerosol weighted altitude $z_{aer}$ <=1.75 km and red for $z_{aer}$ >1.75 km. $N_{PBL}$, $N_{FT}$ are respectively the number of PBL only and PBL plus FT observations, while p50th, p90th are respectively the median, $90^{th}$ percentile of the AOD distribution for both altitude range.

## 7    Conclusions

In conclusion, this study complements several publications (Huang et al., 2010; Sicard et al., 2016) showing that a micropulse lidar is capable of characterizing the variability of the optical properties of aerosols (AOD, vertical profile of the backscatter ratio) at a remote site such as a measuring station in Siberia. In this work, 540 vertical profiles can be used to characterize aerosol sources in Siberia, i.e. a number 7 times larger than that of the largest lidar database used to date for Siberia (Samoilova et al., 2012). A total of 300 daytime and 240 nighttime profiles of backscatter ratio and AODs have been retrieved over periods of 30 min, after a careful calibration factor analysis. Lidar ratio and AODs are constrained with sun-photometer AOD for 70%

of the daytime lidar measurements, while 26% of the nighttime lidar ratio and AOD greater than 0.04 are constrained by direct lidar measurements at an altitude greater than 7.5 km and where a low aerosol concentration is found. It was complemented by an aerosol source apportionment using the Lagrangian FLEXPART model in order to determine the lidar ratio of the remaining 48% of the lidar data. FLEXPART simulations are done with an aerosol tracer and aerosol removal processes for five potential

sources of aerosol emissions. Comparisons of vertical profiles of the backscatter ratio and AOD at 808 nm with sunphotometer AOD and satellite observations show that (1) our approach to attribute an aerosol type to Tomsk lidar observations is validated using satellite observations for 3 case studies, (2) the AOD and lidar ratio calculated for the Tomsk lidar observations are comparable to the sun-photometer daily AOD variability in Tomsk and satellite AOD in the source regions identified by the FLEXPART analysis

According to the analysis of aerosol sources, the occurrence of layers linked to natural emissions (vegetation, forest fires and dust) is high (56%), but anthropogenic emissions still contribute to 44% of the detected layers (1/3 from flaring and 2/3 from urban emissions). The frequency of dust events is very low (5%). When only looking at AOD > 0.1, contributions from Taiga emissions, forest fires and urban pollution become equivalent (25%), while those from flaring and dust are lower (10%-13%). A major advantage of lidar data in AOD climatological studies is the opportunity to discuss the contribution of different altitude

ranges to the large AOD. For example, aerosols related to the urban and flaring emissions remain confined below 2.5 km, while aerosols from dust events are mainly observed above 2.5 km. Aerosols from forest fire emissions are on the opposite observed both within and above the PBL.

*Code availability.* The FLEXPART code version 9.2 was downloaded from the FLEXPART wiki homepage (https://www.flexpart.eu/downloads).

*Data availability.* The CIMEL lidar 372 processed data (AOD, bacscatter ratio) are available on the LATMOS data server and can be

provided on request. The 18-month calibrated lidar data for Tomsk are available on the AERIS infrastructure (http://www.aeris-data.fr). The daily MODIS and VIIRS information from the fires were provided by LANCE FIRMS operated by NASA/GSFC/EOSDIS and are available at https://firms.modaps.eosdis.nasa.gov/download/.

The ECLIPSEv4 data set is available at http://www.iiasa.ac.at/web/home/research/researchPrograms/air/ECLIPSEv4a.html. Level 3 gridded MODIS aerosol parameter data collection 6 were provided in hdf format by ftp://ladsweb.nascom.nasa.gov. The sunphotometer data for

TOMSK have been downloaded from the AERONET data base (https://aeronet.gsfc.nasa.gov). CALIOP level L1 and L2 data have been downloaded from the ICARE date base (http://www.icare.univ-lille1.fr). The AERIS infrastructure (http://www.aeris-data.fr) provided the access to the IASI CO data. Meteorological Analysis are available at ECMWF (http://www.ecmwf.int)

## Appendix A: Lidar aerosol optical depth retrieval

In this appendix, the aerosol optical parameters derived from a backscatter lidar are more precisely described. A backscatter lidar measures the range corrected lidar signal, $P_\lambda(z)$, at range z, which can be related to $\beta_\lambda(z)$ by the following equation:

$$P_\lambda(z) = K_\lambda(\beta_{\lambda,m}(z) + \beta_{\lambda,a}(z)).T_{\lambda,m}(z)^2.T_{\lambda,a}(z)^2. \tag{A1}$$

where $K_\lambda$ is the range independent calibration coefficient of the lidar system, $T^2$ is the two-way transmittance due to any scattering (or absorbing) species along the optical path between the scattering volume at range $z$ and the ground, and $\beta_\lambda$ are the total volume backscatter coefficient at wavelength $\lambda$ with the subscripts m and a specifying, respectively, molecular, and aerosol contributions to the scattering process. For the sake of readability of the text, the reference to $\lambda$ is now omitted. The two-way transmittance for any constituent, x, is

$$T_x^2(z) = \exp(-2\tau_x(z)) = \exp(-2\int_0^z \alpha_x(z')dz'). \tag{A2}$$

where $\tau_x(z)$ specifies the optical depth and $\alpha_x(z)$ is the volume extinction coefficient. Molecular contribution can be estimated with a good accuracy using a molecular density model from ECMWF analysis. When the aerosol contribution is negligible at a range $z_r$ in the free troposphere ($\beta_a(z_r) << \beta_m(z_r)$) and when $\tau_a(z_r) < 0.05$ , one can obtain the lidar system constant $K$

$$K = \frac{P(z_r)}{\beta_m(z_r).T_a^2(z_r).T_m^2(z_r)} \approx \frac{P(z_r)}{\beta_m(z_r).T_m^2(z_r)} \tag{A3}$$

If we divide $P(z)$ by this value and normalize to the Rayleigh contribution, we obtain the attenuated backscatter ratio, $R_{att}(z)$, given by:

$$R_{att}(z) = \frac{P(z)}{K\beta_m(z).T_m^2(z)} = (1 + \frac{\beta_a(z)}{\beta_m(z)}).T_a^2(z) \tag{A4}$$

When $\tau_a(z_r)$ is no longer negligible, the backscatter ratio is obtained using the Fernald backward inversion and assuming a range independent value of the aerosol lidar ratio S (Fernald, 1984):

$$R(z) = \frac{P(z)\exp[-2(S - \frac{8\pi}{3})\int_{z_r}^z \beta_m(z)dz]}{\frac{P(z_r)}{R(z_r)} - 2S\int_{z_r}^z P(z)\exp[-2(S - \frac{8\pi}{3})\int_{z_r}^z \beta_m(z')dz']dz} \tag{A5}$$

$$= \frac{R_{att}(z)\beta_m(z).T_m^2(z)\exp[-2(S - \frac{8\pi}{3})\int_{z_r}^z \beta_m(z)dz]}{\beta_m(z_r).T_m^2(z_r).T_a^2(z_r) - 2S\int_{z_r}^z R_{att}(z)\beta_m(z).T_m^2(z)\exp[-2(S - \frac{8\pi}{3})\int_{z_r}^z \beta_m(z')dz']dz} \tag{A6}$$

The assumption of a range independent aerosol lidar ratio is often not valid (Burton et al., 2012) but it is a well known method to compute the extinction profile for a single wavelength lidar with no independent measurement of the extinction profile (i.e. with a Raman or a High Spectral Resolution Lidar channel). The error remains weak provided that two different aerosol layers with similar contribution to the AOD are not simultaneously present. The two-way aerosol transmittance in A6 is obtained from an independent AOD daytime measurement or the nighttime attenuated backscatter ratio (see A4) if the aerosol

contribution is less than 10% at $z_r$ (i.e. an AOD error of the order of 0.05). When neither of the two previous conditions are met, then $T_a^2$ is obtained by up to 6 iterations of A6. Independent measurements of AOD or nighttime $R_{att}(z_r)$ can also be used to obtain the integrated lidar ratio S using an iterative calculation where an initial value $S_{808} = 60$ sr is assumed to calculate $R(z)$:

$$S = \frac{AOD}{\int_0^{z_r}(R(z)-1)\beta_m(z)dz} = \frac{-\frac{1}{2}\log(R_{att}(z_r))}{\int_0^{z_r}(R(z)-1)\beta_m(z)dz} \tag{A7}$$

## Appendix B:  CALIOP Depolarization ratio analysis

When a linear polarized laser beam is emitted, depolarization related to backscattering in the atmosphere can be measured by a receiving lidar system with an optical selection of the parallel- and cross-polarized signal.The backscatter ratios, R, for perpendicular- and parallel-polarized light are defined as

$$R_\perp(z) = 1 + \frac{\beta_{\perp,a}(z)}{\beta_{\perp,m}(z)} = \frac{R_{att\perp}(z)(1+\delta_m)}{\delta_m T_a^2(z)} \quad R_\parallel(z) = 1 + \frac{\beta_{\parallel,a}(z)}{\beta_{\parallel,m}(z)} = \frac{(R_{att}(z)-R_{att\perp}(z))(1+\delta_m)}{T_a^2(z)} \tag{B1}$$

where $\delta_m = \frac{\beta_{\perp,m}}{\beta_{\parallel,m}}$ is the Rayleigh depolarization, the wavelength dependency of which can be found in Bucholtz (1995), e.g. $\delta_m$=0.015 at 532 nm. The ratio of the aerosol cross- to parallel-polarized backscatter coefficient is called the aerosol depolarization ratio, $\delta_a$, given by:

$$\delta_a(z) = \frac{\beta_{\perp,a}(z)}{\beta_{\parallel,a}(z)} = \frac{R_\perp(z)-1}{R_\parallel(z)-1}.\delta_m = \frac{R(z)\delta(z)(1+\delta_m)-\delta_m}{R(z)(1-\delta(z))(1+\delta_m)-1} \tag{B2}$$

where $\delta(z) = \frac{R_{att\perp}(z)}{R_{att}(z)}$ is the total depolarization ratio. The total depolarization ratio $\delta$ has the advantage of being less unstable when the aerosol layer is weak and it is also less dependent on instrumental parameters (Cairo et al., 1999). The aerosol depolarization being strongly dependent on the accuracy of $R_{532}(z)$, we do not calculate this ratio is $R_{532}(z) < 1.75$

*Author contributions.*  G. Ancellet, J. Pelon and V. Mariage designed the lidar data processing methodology. I. Penner, Y. Balin and S. Nasanov designed and carried out the lidar measurement program in Tomsk. G. Ancellet carried out the FLEXPART analysis. J.C. Raut
provided the aerosol source inventories. A. Zabukovec carried out the case study satellite data analysis of Section 5. G. Ancellet and J. Pelon wrote the manuscript with contribution from all co-authors.

*Competing interests.*  No competing interests are present

*Acknowledgements.*  This work was supported by the CNES EECLAT project, the iCUPE H2020 project and the Chantier Arctique Français (PARCS). The work was also supported in part by Ministry of Education of Science of RF (Agreement No. 14.616.21.0104, unique identifier

RFMEFI61618X0104) We thank the European Centre for Medium Range Weather Forecasts (ECMWF) for the provision of ERA-Interim reanalysis data and the FLEXPART development team for the provision of the FLEXPART 9.2 model version used in this publication. The authors thank the AERIS infrastructure and NASA/GSFC for providing the satellite data used in this paper (CO, AOD, CALIOP and fire FRP).

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
