# Peer review of "Aerosol monitoring in Siberia using an 808 nm automatic compact lidar"

_Atmospheric Measurement Techniques, 2018_

## Referee Comment (RC1) · Anonymous Referee #2 · 25 Sep 2018

The paper deals with the analysis of a micropulse lidar measurements at 808 nm near the city of Tomsk (Siberia) for the period April 2015 – September 2016. The papers uses an hybrid methodology of air-mass analysis and aerosol sources to inferr aerosol lidar ratio. The model used for air-mass backward computation is the well-known FLEXPART model. Authors evaluate the consistency or their methodology using MODIS and CALIPSO spaceborne measurements for three particular months.

The study of aerosol vertical distribution is very important generally and particularly for Siberia where there are not many studies, as the authors brilliantly highlight in the introduction section. The topic is sound and of interest for its publication in Atmospheric measurement Techniques. However, the current paper need further revisions before its final publication in Atmospheric Measurement Techniques.

[Figure]

**MAJOR CONCERNS**

FLEXPART is mainly developed for the analysis of air-masses, both in backward and forward models. The authors propose FLEXPART use in combination with hyphotesis of aerosol sources over different regions. However, I wonder why not using more sophisticated models that already include aerosol modules. An example could be the NASA Goddard Earth Observing System Model, Version 5 (GEOS-5, https://gmao.gsfc.nasa.gov/GEOS/) or other from ECWMF. These model already include aerosol emissions and depositions.

It is not clear the novelty proposed about the analysis of micropulse lidar data. Actually, authors claim in the conclusion section that they analyze extinction and backscattering measurements, which is not true because a micropulse system needs the assumption of an aerosol lidar ratio. This give a general ambiguety to the scientific discussion. The authors must clarify the novelty they propose. For example, when FLEXPART is used for obtaining lidar ratios, Are consistent with the cases when sun-photometer are also available? Also, nighttime retrievals of lidar ratio are not clear. Appendices 'A' and 'B' do not provide any novelty to what is already known in lidar aerosol optical depth retrieval and in CALIOP depolarization ratio computation. Finally, the evaluation or the new methodology would need of correlative measurements of Raman/HSRL systems. If these measurements are not available, at least reference them in the text.

The authors selected three study cases and three months to present their analysis and the links with satellite observations. But to me it is not clear why these cases are representative.

**MINOR CONCERNS**

Pag. 9: Authors say that Russia emissions are not well-known but they use ECLIPSEv4 dataset for emissions. That seems a contradiction. Please clarify

Pag. 2, Line 13: EARLINET network posses more sophisticated instruments such as

Raman lidar, which provide further information on aerosol vertical-profiles. The authors should include this in the introduction. Also, there are many measurements of Raman lidars in North America, Asia and Latin America. The authors should not ignore that.

Pag. 2, Line 15: Please define what is CIF.

Pag. 2, Line 26: The best estimates of Angström exponent are provided by MISR satellite, not by MODIS. Please correct.

Pag. 2, Lines 29-30: CALIPSO does not provide direct estimates of aerosol extinction. CALIPSO is backscattering lidar. Please correct.

Pag. 3, Line 8: Why not complementing your study with VIIRS satellite?

Section 2.1: Is the lidar system operating continously?

Section 2.1: What is the vertical resolution of your lidar system?

Pag. 4, Line 33: Please, provide references for ERA-Interim.

Pag. 5: Please provide a better explanation of your iterative method for computing lidar ratio. Why starting at 60 sr? What happen with dust cases?

Pag. 5: Please give a complete definitions of the variables in Line 2 and in equation 1.

Pag. 7, Line 6: It is difficult to understand how you obtain the final accuracy on calibration factor. Please give a better description.

Pag. 8 and 9: Please provide the link for MODIS and VIIRS data.

Pag. 9: Please, provide a link for ECLIPSEv4 database.

Pag. 10, Line 1: This statement is incorrect. Compation of AOD requires vertical-profile of lidar ratio which is possible with Raman and HSRL systems but not with micropulse lidars. Your approach assumes constant lidar ratio. Please correct.

Pag. 11, Line 3: This statement is incorrect. Currently, it is possible to obtain AOD during the night by star and moon photometry (see ACTRIS project for example). Please correct.

Figure 5: Please, explain better how you compute your nighttime AOD. Particularly, how do you obtain nighttime lidar ratios.

Figure 7: It is difficult to follow how was the aerosol load during the different periods you claim. Why not adding AOD temporal evolution?

Section 5.1.1 must be strenghten. The scientific discussion is poor. It is not clear the large diurnal variability in lidar values for Case C. Are each of the cases selected illustrative of the different atmospheric conditions over the study area?

Pag 18, Line 6: It is not clear if you use standard CALIPSO data or your own computations, particularly for depolarization measurements.

Pag 20, Line 13: The PBL is a region in your profile not a source of aerosol emissions. Please correct this.
* * *

---

## Referee Comment (RC2) · Anonymous Referee #1 · 10 Oct 2018

The submitted manuscript shows the results of a lidar measurement campaign that took place in Siberia, near the city of Tomsk, from April 2015 to September 2016. Despite the measurements of the vertical profile of the aerosol geometrical, optical and microphysical properties are extremely precious in this wild and remote region of the world, the manuscript is still not ready for publication, suffering probably from hasty writing, as some sentences are approximate (see next sections). However, I am sure that the authors have will brilliantly address the issues raised below.

Major Issues:

The described measurement technique is not introducing any innovative aspect. Since Klett (1981) and Fernald (1984) papers, tens of research articles were published about elastic lidar signal inversion, together with their pros and cons. For example, the tech-

nique described in Section 2 is operational (with some minor differences) since 1999 in the NASA MPLNET lidar network. However, all those different inversion methods using either the retrieved sunphotometer AOD to constrain LR or taking it directly from a model as FLEXPART (or a combination of both), still assume that the LR is constant over the atmospheric column. This might introduce large bias and uncertainties, especially when co-exist different aerosol layers at different altitudes. For those reasons, the manuscript is unbalanced as a large part of it is dedicated to describe the retrieval technique (including Appendix A and B)

The instrument wavelength (808nm) might be more appropriate to study clouds than aerosols. The molecular signal at this wavelength is about 5 times less than at 532 nm and about 26 times less than 355nm making calibration very difficult (impossible during daytime as showed in Fig. 1). Moreover, the backscattering from the sub-micron part of the aerosol spectrum is almost negligible.

FLEXPART model is used to speciate the aerosol layers and quantitatively assess the columnar LR to be used in the inversion. However, many parameters are assumed without giving convincing explanations., i. e. A and B. How the results change if, for example, the number of released particles changes and also the altitude?

I suggest to put more emphasis on characterizing the source origins, transport processes, and vertical distributions of the aerosol layers on the region, possibly integrating the lidar observation with in-situ measurements, if available.

Specific Comments:

Line 1, Abstract: The word climatology is not appropriate considering the total number of measurement.

Line 8-9, Abstract: "it was complemented..." please rephrase as it is not clear. Do the authors mean ancillary ?

Line 9-10, Abstract, The sentence is unclear. What exactly is compared? attenuated

backscattering with CALIPSO? or What? and what is it compared with MODIS and IASI data?

Line 10, Pag. 2 the term "Radiative Forcing" is misused. I would change it into "Radiative Effects"

Line 13, Pag. 3 "continuous measurements of clouds and aerosols" again, this sentence lacks of precision. Please specify what it is measured.

Line 6, Pag.4 "counts/s"

Line 16 pag 4. How much is it the lidar blind region ? (overlap 0%).

Line 31 pag 4 supposing clear air at 2-4 km altitude is very risky

Figure 1 upper plot labels are very small and can't be read

Pag. 5 bottom: fire is not a good choice, I would say biomass burning

Line 8, Pag. 6 why 35%? any reference?

Line 1 Pag. 7 how much is the lidar sensible to the thermal stability?

Line 30 Pag. 9. How is retrieved the AOD at 808nm from 870nm?

Line 1 pag 10. how much is it the integration time to get a good molecular signal ? 30 mins are enough?

Figure 8. is 3.3 and 16.1 the time? it is pretty uncommon way...and the caption should be more detailed

Figure 9,10. same as Fig. 8

Line 11 Pag. 19. The link is broken

It is missing a discussion why the selected cases are representative of the region

Line 13 Pag,. 22 Micropulse lidar in Sicard et al., 2016 is more suitable to study aerosol

variability being at 532nm

---

## Author Comment (AC1) · 14 Nov 2018

[a4paper,11pt]report xcolor

**Answer to Anonymous Referee 2**

The authors gratefully acknowledge the critical review of the manuscript and the remarks have been carefully taken into account in the corrected version. We hope the new version now emphasizes that a micropulse lidar backscatter can provide useful information about the AOD seasonal variability and the aerosol vertical structure, even though Raman/HSRL lidar remains more powerful tools to obtain a direct extinction vertical profile. The modifications are shown in red within the new version of the manuscript (see supplementary document).

**MAJOR CONCERNS**

FLEXPART is mainly developed for the analysis of air-masses, both in backward and forward models. The authors propose FLEXPART use in combination with hyphotesis of aerosol sources over different regions. However, I wonder why not using more sophisticated models that already include aerosol modules. An example could be the NASA Goddard Earth Observing System Model, Version 5 (GEOS-5, https://gmao.gsfc.nasa.gov/GEOS/) or other from ECWMF. These model already include aerosol emissions and depositions.

FLEXPART is a good trade-off between simple back-trajectory analysis from known aerosol source regions and 3D aerosol model simulations. FLEXPART is well adapted to represent air mass transport between a source and a measurement site (see Stohl et al. 2002 ). NASA GEOS5, ECMWF, NAAPS/NRL are indeed very useful if a quantitative assessment of the aerosol load is needed (e.g. for radiative transfer calculation). They generally provides aerosol mapping with a resolution coarser than the data from a single observing station. Furthermore in our study we only need to identify the aerosol type in order to select the appropriate aerosol optical properties to be used in the lidar data processing. A better justification is added p.10 l.7-l.13 in the introduction of section 3.

It is not clear the novelty proposed about the analysis of micropulse lidar data. Actually, authors claim in the conclusion section that they analyze extinction and backscattering measurements, which is not true because a micropulse system needs the assumption of an aerosol lidar ratio. This give a general ambiguety to the scientific discussion. The authors must clarify the novelty they propose.

We agree that the novelty of the methodology was not well described and we tried to improve this throughout the paper

First the old section 4 was thoroughly revised by:

- splitting the old section 4 into the methodology part (now section 2.3) and the
discussion of the results in section 4 in order to clarify the contribution to lidar processing methodology and to new results about aerosol characterization in Siberia,

- adding a new figure (Fig. 4) with the lidar data processing flow chart in order to show that a non-standard methodology is indeed needed when dealing with a 808 nm lidar (no molecular signal during the day) and with a long 18-month period where measurement conditions changed (nighttime or daytime, cloudy or clear sky in the 4-7 km altitude range)

- adding a new paragraph at the beginning of section 2.3 (p.8 l.13 to p.9 l.10) to clarify that 4 different measurement conditions must be accounted for: (1) two cases with constrained lidar ratios with independent Ta2 measurements (daytime data with sun-photometer, nighttime with measurements of Ta2 above 7.5 km), (2) two cases with lidar ratios taken from a lookup table built in this work using the constrained lidar ratios and the FLEXPART analysis.

Second reference to extinction profile was removed in the conclusion.

Third, a backscatter lidar is of course limited in the assessment of AOD compared to Raman or HSRL lidar. We already recognized this in the initial manuscript but probably not enough. This is now stated in the introduction (see p.2 l.19-l.24 and p.3 l.20-l.24). Nevertheless, we disagree with the view that nothing can be done with a backscatter lidar in terms of AOD measurements. Various methodologies were used in previous works (external AOD measurement with a sun-photometer, aerosol type climatology, e.g. in the CALIOP algorithm) to provide indirect estimate of the AOD. The purpose of this paper is to propose a methodology combining direct (nighttime) and indirect (daytime) AOD retrieval corresponding to our specific measurement conditions in Siberia. We agree it is less straightforward than dealing with direct extinction profile measurements using a Raman lidar, but our first results, i.e. the comparison with sunphotometer

AOD (Fig. 8 and 9), and the variability of the lidar ratio for the different aerosol type (Table 1), look promising.

For example, when FLEXPART is used for obtaining lidar ratios, Are consistent with the cases when sun-photometer are also available?

The use of FLEXPART has been probably misunderstood. It is now clarified in the section 2.3 (see p.8 l.13 to p.9 l.10) and Fig. 4. FLEXPART is used to:

- build the lidar ratio lookup table (Table 1) using the sun-photometer or nighttime upper troposphere molecular signal from the calibrated lidar,

- select the lidar ratio from the lookup table for the cases where the lidar ratio cannot be constrained (daytime without sun-photometer or nighttime without a molecular signal detectable in the upper troposphere).

The consistency of the AOD using the lookup table with the sunphotometer AOD are indeed checked for the case studies shown in section 5.1 (see Fig. 9).

Also, nighttime retrievals of lidar ratio are not clear. Appendices 'A' and 'B' do not provide any novelty to what is already known in lidar aerosol optical depth retrieval and in CALIOP depolarization ratio computation.

Yes we agree the Appendix are not meant to develop the innovative aspect but only to provide the basic lidar data processing for readers with no lidar background. The innovative aspect is now discussed in section 2.2 (lidar calibration) and section 2.3 (AOD retrieval combining direct and indirect method), while the methodology is explained in the new figure 4.

Regarding the nighttime retrieval, it is probably the point less commonly found in the lidar data processing literature and it is now better explained that the key point is that a calibrated lidar provide a direct Ta2 measurement using the molecular signal in the free troposphere during the night. A constrained integrated lidar ratio can then be also

obtained during the night. Another approach would be to use a lunar photometer, but it was not possible to install one in Tomsk. It is better explained in section 2.3 and in Fig. 4.

Finally, the evaluation or the new methodology would need of correlative measurements of Raman/HSRL systems. If these measurements are not available, at least reference them in the text.

Yes direct comparisons between a Raman lidar and a micropulse data processing using our approach would be a nice follow-up of this work. Unfortunately it was not possible for the very remote place where the lidar was deployed. Nevertheless references to existing Raman lidar contribution are added in the introduction (p.2 l.13-l.24). Is is also discussed when the lidar ratio retrieved by our analysis are compared with the lidar ratios found in the existing literature, since the latter are often based on Raman/HSRL lidar (Burton et al. 2012, Hofer et al. 2017) . It is clarified in section 4.2 where the Table 1 shows the lidar ratio variability for different aerosol sources (p.13 l.21-p.14 l.9) and in section 5.1 using the S808 daily variability during the 3 selected episodes (p.18 l.2-l.10, and p.18 l.13-p.19 l.8 and p.19 l.13-p.20 l.2)

The authors selected three study cases and three months to present their analysis and the links with satellite observations. But to me it is not clear why these cases are representative.

We agree with the reviewer. This section was poorly written. It has been significantly changed to clarify the goal and to present the results.

The introduction of section 5 now reads:

" In this section, we focus on the time periods with elevated AOD observed by the AERONET network above Tomsk in order to (1) compare the results of our AOD analysis with AERONET values during 48 h around the selected lidar profiles and with satellite data (MODIS or CALIOP) (2) identify the likely aerosol sources derived from the FLEXPART analysis with satellite observations (MODIS, IASI, CALIOP) in the source

areas . Looking at Fig. 8, there are 5 time periods with sun-photometer AOD > 0.25: mid-may 2015, end of may 2015, April 2016, mid-June 2016 and end of September 2016. We do not have enough lidar data for mi-June 2016. The end of September 2016 and mid-June 2015 cases both correspond to forest fire events, while end of may 2015 and April-2016 correspond to urban, flaring and dust emissions according to our FLEXPART analysis. Therefore the three time periods corresponding to periods A, B, C of Fig. 8 are analyzed in this section. The section 5.1 presents the daily variability of the lidar backscatter profiles and sun-photometer AOD, while the section 5.2 presents the analysis of satellite observations."

**MINOR CONCERNS**

Pag. 9: Authors say that Russia emissions are not well-known but they use ECLIPSEv4 dataset for emissions. That seems a contradiction. Please clarify

The ECLIPSEv4 dataset is only meant to identify the location of the flaring emission in section 3. Underestimation of the ECLIPSEv4 emission factor which is reported in the literature, will not strongly bias the assessment of the flaring source location. This underestimate would be detrimental only if the emission factor was used to calculate the aerosol concentrations and related aerosol optical properties. This does not apply to our work with ECLIPSEv4.

Pag. 2, Line 13: EARLINET network posses more sophisticated instruments such as Raman lidar, which provide further information on aerosol vertical-profiles. The authors should include this in the introduction. Also, there are many measurements of Raman lidars in North America, Asia and Latin America. The authors should not ignore that.

This was recognized in the previous introduction but not enough. More references to Raman lidar aerosol monitoring are added (see p.2 l.13-l.16).

Pag. 2, Line 15: Please define what is CIF.

Done

Pag. 2, Line 26: The best estimates of Angström exponent are provided by MISR satellite, not by MODIS. Please correct.

Done. Reference to MISR is added in the introduction (p.3 l.5-l.7). In the paper the AE is always derived from the AERONET sunphotometer observations and not from MODIS.

Pag. 2, Lines 29-30: CALIPSO does not provide direct estimates of aerosol extinction. CALIPSO is backscattering lidar. Please correct.

Sentence changed. CALIPSO still provides indirect aerosol extinction profiles (see Winker et al. 2013, Omar et al. 2009). See p.3. l.10.

Pag. 3, Line 8: Why not complementing your study with VIIRS satellite?

It is included in the biomass burning aerosol source analysis (see section 3.2 p.11 l.17-l.23)

Section 2.1: Is the lidar system operating continously?

Yes see introduction of section 2 (p.4 l.1), but for the aerosol analysis we must use a 30 min. time averaging with no cloud between 0-4 km (p.4 l.27-l.33)

Section 2.1: What is the vertical resolution of your lidar system?

Vertical resolution is 15 m (see p.4 l.23) and additional 150 m vertical filtering for the assessment of the molecular signal (sentence added p.5 l.24).

Pag. 4, Line 33: Please, provide references for ERA-Interim. Done

Pag. 5: Please provide a better explanation of your iterative method for computing lidar ratio. Why starting at 60 sr? What happen with dust cases?

Thank you for suggesting to clarify this. The following text is added in section 4.2 p.13 l.19: "Starting with the largest expected lidar ratio allows a fast convergence towards the true value (e.g. Young95). Thirteen $S_{808} < 45$ sr could be retrieved with this method

out of the 15 FLEXPART dust cases even though iteration starts with 60 sr."

Pag. 5: Please give a complete definitions of the variables in Line 2 and in equation 1.

I am not sure what the reviewer meant, variables are actually defined p.5 l.26 and p.6 l.3-l.4.

Pag. 7, Line 6: It is difficult to understand how you obtain the final accuracy on calibration factor. Please give a better description.

We agree it was not so clear. The text now reads (p.8 l.1):

"To estimate our error on K values for non-optimal conditions (red points in Fig. 3), a good proxy is the difference between two optimal calibration factors derived for two observations made with a time difference $< 1$ day. Changes of $K_{op}$t for such a short time period cannot be expected when aiming at calibration of daytime observations with a nighttime calibrated profiles. There are 23 pairs of $K_{opt}$ values with a 1-day time difference and the standard deviation of their difference, $DK_{opt}$, is 2.5 $10^4$. Such a variability is then a limiting factor in our ability to calibrate the lidar for daytime observations or nighttime conditions with either AOD$>$0.06 or clouds between 4 and 7.5 km. The corresponding accuracy on the calibration factor K is then of the order of 8% (2.5 $10^4$ / 3 $10^5$)"

Pag. 8 and 9: Please provide the link for MODIS and VIIRS data. Done in section data availability

Pag. 9: Please, provide a link for ECLIPSEv4 database.

Done in section data availability

Pag. 10, Line 1: This statement is incorrect. Compation of AOD requires vertical-profile of lidar ratio which is possible with Raman and HSRL systems but not with micropulse lidars. Your approach assumes constant lidar ratio. Please correct.

We disagree with the reviewer. It is possible to derived directly the AOD during the night

if the lidar is well calibrated and if the molecular signal can be detected in the upper troposphere. This is explained in section 2.3 and it is the backbone of our independent estimate of Ta2 during the night. Since it was not clear enough, now section 2.3 details the methodology and a specific section (section 4.1) presents the results obtained with the direct AOD measurements (see also answer to major comments).

Pag. 11, Line 3: This statement is incorrect. Currently, it is possible to obtain AOD during the night by star and moon photometry (see ACTRIS project for example). Please correct.

We agree. This possible alternative is included (sentence p.13 l.11) even though no lunar photometry was available in Siberia during our lidar measurement period.

Figure 5: Please, explain better how you compute your nighttime AOD. Particularly, how do you obtain nighttime lidar ratios.

This has been clarified (see answer to major comments)

Figure 7: It is difficult to follow how was the aerosol load during the different periods you claim. Why not adding AOD temporal evolution?

We thank the reviewer for suggesting this AOD comparison when validating the AOD lidar retrieval for the 3 case studies. A new figure is added (Fig.9) to discuss the AOD diurnal variability measured by the sunphotometer and the lidar. More lidar vertical profiles are also reported in Fig. 10 to 12. It is now discussed in section 5. 1 (see major comments)

Section 5.1.1 must be strenghten. The scientific discussion is poor. It is not clear the large diurnal variability in lidar values for Case C. Are each of the cases selected illustrative of the different atmospheric conditions over the study area?

We fully agree it was a weak point in the initial version. Section 5.1 has been strongly modified (see answer to major comments)

Pag 18, Line 6: It is not clear if you use standard CALIPSO data or your own computations,
particularly for depolarization measurements.

We use our own computation for the CALIPSO backscatter ratio profile, AOD and depolarization ratio, but it is based on CALIOP Level 1 attenuated backscatter and level 2 aerosol and cloud data products. The IR imager brightness temperatures are also used to check the aerosol/cloud discrimination. Lidar ratio is usually based on the CALIOP L2 data products. It is explained in section 5.2.1 p.21 l.12-p.22 l.8 More details can be found in Ancellet et al. 2014.

Pag 20, Line 13: The PBL is a region in your profile not a source of aerosol emissions. Please correct this.

Done

---

## Author Comment (AC2) · 14 Nov 2018

**Answer to Anonymous Referee 1**

The authors gratefully acknowledge the critical review of the manuscript and the remarks have been carefully taken into account in the corrected version. We hope the new version now emphasizes that the added value is not only related to the analysis of aerosol source identification and seasonal variability in Siberia but also related to the proposed methodology to analyze a 808 nm backscatter lidar. The modifications are shown in red within the new version of the manuscript (see supplementary document)

**Major Issues:**

The described measurement technique is not introducing any innovative aspect. Since Klett

(1981) and Fernald (1984) papers, tens of research articles were published about elastic lidar signal inversion, together with their pros and cons. For example, the technique described in Section 2 is operational (with some minor differences) since 1999 in the NASA MPLNET lidar network. However, all those different inversion methods using either the retrieved sunphotometer AOD to constrain LR or taking it directly from a model as FLEXPART (or a combination of both), still assume that the LR is constant over the atmospheric column. This might introduce large bias and uncertainties, especially when co-exist different aerosol layers at different altitudes.

We agree with the reviewer that Raman and HSRL lidar do provide better extinction vertical profile retrieval when multiple layers with different aerosol type are present. The purpose of this paper is to assess if an automatic backscatter lidar is still valuable in a remote place such as Siberia, with the additional difficulty that it is running in the IR at 808 nm to reduce cost and meet the eye-safe requirement. Therefore the novelty of the paper is threefold (1) propose a methodology for the retrieval of aerosol optical depth (AOD) based on a CALIBRATED IR lidar in addition to well known techniques based on the use of sun-photometer (section 2.3) (2) assess the retrieved AOD by comparison with AERONET AOD and discuss the variability of the corresponding lidar ratio (section 4.1, 4.2 and 5) (3) analyze daily measurements over two years (18 months of effective measurements) to obtain a statistically significant data set of backscatter vertical profile and total AOD in order to discuss the aerosol sources and variability (section 4.3 and 6).

Regarding the question of multiple layers with different lidar ratio, it is indeed the main drawback when using a backscatter lidar. It is now better recognized in the introduction. However we believe that our results show that the bias in the AOD retrieval remains limited for our database. First the time variation of the vertical aerosol vertical structure (Fig. 7) shows that the main contribution to the bascckscatter ratio is within the planetary boundary layer (PBL) reducing the bias due to the assumption of a lidar ratio constant with altitude. Second the comparisons of the lidar AOD with the sunphotometer AOD in

section 5.1 show a good agreement (Fig. 9) even though some profiles show multiple layers in Fig. 10 to 12.

We have clarified these different points:

- in the introduction, it is stated that Raman lidar will always be better suited for the aerosol extinction profile retrieval, but that backscatter lidar has been used in the past for aerosol characterization and must be characterized for measurements of AOD and backscatter ratio in Siberia (p.2 l.13-l.24 and p.3 l.20-l.24),

- the originality of the lidar data processing is better explained (new figure 4, new section 2.3 p.8 to p.10, better explanation of the rationale for lidar calibration in section 2) showing that the approach goes beyond the simple use of Klett inversion technique (nighttime direct retrieval, lidar ratio retrieval from a lookup table)

- the section 4 now includes only results of nighttime direct retrieval of the AOD (section 4.1) and discussion of the integrated lidar ratio retrieval which can be used to build the lidar ratio lookup table to be used with the FLEXPART analysis (section 4.2).

- the section 5 now includes a discussion of the AOD retrieval error by a comparison with the daily sunphotometer AOD for the 3 selected case studies, i.e. the main aerosol sources observed in Tomsk: biomass burning, dust, anthropogenic sources (new Fig. 9, more profiles in Fig. 10 to 12, discussion of lidar AOD p.17 to 19)

- the section 5 now includes a discussion of the lidar ratio $S_{808}$ variability when the air mass transport changes for the 3 case studies (discussion of $S_{808}$ p.17 to 20 ).
For those reasons, the manuscript is unbalanced as a large part of it is dedicated to describe the retrieval technique (including Appendix A and B)

We believe that the paper is not balanced towards the description of the measurement technique as a large part of the paper is devoted to the analysis of the variability of aerosol sources and to the comparison with sun-photometer and satellite data for three "typical" aerosol sources encountered during the analysis (14 pages for the discussion of the results compared to 9 pages for the instrument description and presentation of the methodology. Old section 4 is now divided in two parts to distinguish the methodology description (new section 2.3) and the discussion of the AOD and backscatter ratio results (new section 4). We agree however that the scientific discussion about the case studies in section 5 was poor and could be better detailed. It is now improved in the new version: new discussion of sunphotometer and lidar AOD comparison, new discussion of lidar ratio daily variability, more lidar vertical profiles are included in the analysis (Fig.10 to 12). A new introduction to section 5 is added (p.15 l.19-p.16 l.4 see hereafter) to clarify the goals, while the text in section 5.1 was significantly changed to emphasize the contribution to the AOD retrieval validation.

The introduction of section 5 now reads:

" In this section, we focus on the time periods with elevated AOD observed by the AERONET network above Tomsk in order to (1) compare the results of our AOD analysis with AERONET values during 48 h around the selected lidar profiles and with satellite data (MODIS or CALIOP) (2) identify the likely aerosol sources derived from the FLEXPART analysis with satellite observations (MODIS, IASI, CALIOP) in the source areas . Looking at Fig. 8, there are 5 time periods with sun-photometer AOD $> 0.25$: mid-may 2015, end of may 2015, April 2016, mid-June 2016 and end of September 2016. We do not have enough lidar data for mi-June 2016. The end of September 2016 and mid-June 2015 cases both correspond to forest fire events, while end of may 2015 and April-2016 correspond to urban, flaring and dust emissions according to our FLEXPART analysis. Therefore the three time periods corresponding to periods A, B,

C of Fig. 8 are analyzed in this section. The section 5.1 presents the daily variability of the lidar backscatter profiles and sun-photometer AOD, while the section 5.2 presents the analysis of satellite observations."

The instrument wavelength (808nm) might be more appropriate to study clouds than aerosols. The molecular signal at this wavelength is about 5 times less than at 532 nm and about 26 times less than 355nm making calibration very difficult (impossible during daytime as showed in Fig. 1). Moreover, the backscattering from the sub-micron part of the aerosol spectrum is almost negligible.

Of course we are aware of the drawbacks when using an IR laser source in a backscatter lidar and especially the lack of molecular signal detection during the day. Nevertheless these instruments are and will be used in monitoring network for cloud and aerosol owing to their cost, their size and their very stable laser transmitter. It is therefore useful to assess the aerosol measurement capabilities. We do not attempt to calibrate the lidar during daytime and 30 min. time integration of nighttime profiles is enough to detect the molecular signal in the upper troposphere (see Fig. 1) and lidar calibration is then possible.

Regarding the sensitivity of the aerosol detection with a 808 nm lidar, the low molecular signal is a strong advantage to identify aerosol layers (layering seen for the case studies discussed in section 5 are good examples of this, see also the time/elevation plot for July 2nd 2015 attached to this document in order to illustrate the capabilities of the lidar to monitor aerosol layering in the free troposphere during the night). Even though a 808 nm lidar is less sensitive to the sub-micronic part of the aerosol spectrum as pointed out by the reviewer, it is balanced by the large sub-micronic aerosol concentration (e.g. see Paris et al. Atmos. Environ. 2009 https://doi.org/10.1016/j.atmosenv.2008.11.032). The rationales for using an IR lidar and the need for assessment of aerosol monitoring using a 808 nm lidar are now better explained in the introduction (see p.2 l.19-l.24).

FLEXPART model is used to speciate the aerosol layers and quantitatively assess the colum-

nar LR to be used in the inversion. However, many parameters are assumed without giving
convincing explanations., i. e. A and B. How the results change if, for example, the number of
released particles changes and also the altitude?

FLEXPART is used to sort the lidar profile according to a potential emission source.
It is a much better approach than using simple back-trajectories. We have chosen an
"aerosol like" tracer i.e. sensitive to dry deposition and scavenging to get a more realis-
tic assessment of an aerosol source for long range transport through cloudy conditions.
The parameters for wet scavenging have been chosen following previous studies using
FLEXPART aerosol tracers (see Stohl 2013, Stohl 2012, Kriestiensen 2016 ). The re-
sults are usually weakly dependent on the number of particles released. It is of course
strongly dependent on the altitude, this is why we select aerosol layer thickness suffi-
ciently broad ($>$1 km) to minimize the sensitivity to strong differential advection, while
being still specific of the airmass origin. I am not sure what kind of sensitivity studies
are suggested by the reviewer. FLEXPART has been validated including aerosol tracer
in many publications, so we do not wish to add more characterization of this tool in our
work. More references to previous FLEXPART work are now given (Stohl et al. 2012,
Kriestensen et al. 2016).

I suggest to put more emphasis on characterizing the source origins, transport processes, and
vertical distributions of the aerosol layers on the region, possibly integrating the lidar observa-
tion with in-situ measurements, if available.

Unfortunately we do not have in-situ observations which can be discussed together
with the lidar profiles, e.g. in section 5 describing the case studies. As stated in the
answer to the first comment, we believe that the paper is now well balanced between
the analysis of the aerosol sources using the lidar data (14 pages) and the presentation
of the lidar data processing technique (9 pages). The paper would be incomplete if the
data are discussed without explaining the procedure to derive the backscatter ratio and
the AOD, especially for 808 nm micropulse lidar where the retrieval is not so straight-
forward. The new version is more balanced in the way proposed by the reviewer by

making a better description and analysis of the lidar data for the three selected case studies (section 5) and splitting the old section 4 in two parts to distinguish the presentation of the methodology (section 2.3) and the discussion of the results (section 4). Along the same lines, the section 5.2 describing the satellite data analysis above the source region identified by FLEXPART is split in two parts to distinguish the description of the satellite data products (section 5.2.1) and the results (section 5.2.2) on the aerosol properties above the source region.

To be more specific the text changes in section 5 are the following:

- discussion of AOD comparison and vertical profiles daily variability: p.17 l.3-l.6 and p.18 l.11-l.12 and p.19 l.9-l.10

- discussion of $S_{808}$ daily variability: p.18 l.2-l.10, and p.18 l.13-p.19 l.8 and p.19 l.13-p.20 l.2

**Specific Comments:**

Line 1, Abstract: The word climatology is not appropriate considering the total number of measurement.

Agreed although it is the first time such a large number of lidar profiles is discussed for Siberia. Climatology is replaced by "seasonal variability".

Line 8-9, Abstract: "it was complemented..." please rephrase as it is not clear. Do the authors mean ancillary ?

Sentence changed: "An aerosol source apportionment using the Lagrangian FLEX-PART model is used in order to determine the lidar ratio of the remaining 48% of the lidar database".

Line 9-10, Abstract, The sentence is unclear. What exactly is compared? Attenuated backscattering with CALIPSO? or What? and what is it compared with MODIS and IASI data?

Agreed. The sentence now reads: "Backscatter ratio vertical profile, aerosol type and $AOD_{808}$ derived from micropulse lidar data are compared with sunphotometer $AOD_{808}$ and satellite observations (CALIOP spaceborne lidar backscatter and extinction profiles, Moderate Resolution Imaging Spectroradiometer (MODIS) AOD550 and Infrared Atmospheric Sounding Interferometer (IASI) CO column) for three case studies corresponding to the main aerosol sources with $AOD_{808} > 0.2$ in Siberia."

Line 10, Pag. 2 the term "Radiative Forcing" is misused. I would change it into "Radiative Effects" Agreed Line 13, Pag. 3 "continuous measurements of clouds and aerosols" again, this sentence lacks of precision. Please specify what it is measured.

Agreed replaced by "measurements of cloud and aerosol backscatter".

Line 6, Pag.4 "counts/s" Changed

Line 16 pag 4. How much is it the lidar blind region ? (overlap 0%).

The truly blind region is of the order of 100 m. Assessment of the AOD error when assuming a constant backscatter ratio below 100 m is added (see p.5 l.10-l.12).

Line 31 pag 4 supposing clear air at 2-4 km altitude is very risky

In fact this assumption is only for the first guess retrieval which is no longer used after the calibration of the daytime profile with the nighttime calibration factor. This sentence was removed as it has caused confusion. The calibration procedure has been clarified in section 2.2 (see p.6 l.1-l.4 and p. 7 l.3-l.5) and figure 4 now describes the lidar processing flowchart.

Figure 1 upper plot labels are very small and can't be read

Agreed Figure 1 upper panel has been changed

Pag. 5 bottom: fire is not a good choice, I would say biomass burning Agreed.

Line 8, Pag. 6 why 35%? any reference?

35% corresponds to the expected range for lidar ratio (35-60 sr) assuming that all the aerosol types can be encountered except the clean marine or dusty marine types. It is now better explained in section 2.2 and references to CALIPSO (Omar et al. 2009) and AERONET (Cattrall et al. 2005) have been added (p.7 l.3-l.5)

Line 1 Pag. 7 how much is the lidar sensible to the thermal stability?

Although the lidar box is thermally controlled, a gradual change of outside temperature remains the main source of gain variability (e.g. detuning of the detection interference filter). It is difficult to provide more direct quantitative results apart from the measured variability of the calibration factor discussed in section 4 and in Fig. 3.

Line 30 Pag. 9. How is retrieved the AOD at 808nm from 870nm?

The AERONET sunphotometer angstrom exponent (AE) measurement is used to estimate the AOD spectral variability between 808 nm and 870 nm. It was explained p.9 l.11-l.13.

Line 1 pag 10. how much is it the integration time to get a good molecular signal ? 30 mins are enough?

Yes 30 min. is enough during nighttime conditions since the statistical noise in the 30-min averaged PR2 is of the order of 7-8% at 9 km (e.g. see Fig. 1). Vertical averaging over 150 m at the reference altitude is also applied to derive the molecular signal to keep the uncertainty on the molecular signal below 3%. It is clarified in section 2.2 p.5 l.24-l.26, when we described the calibration method. Figure 8. is 3.3 and 16.1 the time? it is pretty uncommon way...and the caption should be more detailed

We are not sure what the reviewer meant. Captions of figure 8 to 10 are now more detailed

Line 11 Pag. 19. The link is broken It is missing a discussion why the selected cases are representative of the region

Corrected (l missing at the end of web address)

Line 13 Pag,. 22 Micropulse lidar in Sicard et al., 2016 is more suitable to study aerosol variability being at 532nm

Yes we agree if cost, eye-safe requirements, laser liftetime are not limiting factors in the lidar deployment (see the major issues discussion)

[Figure]

**Supplement:**

[revised manuscript text omitted]

---

## Author Response (AR2)

**Answer to Anonymous Referee 1**

I am happy that the authors addressed all my previous issues. The manuscript is now ready for publication after some minor changes.

Line 10 pag 2. I suggest to broad the discussion with few rows, stressing that aerosol layers can either cool or warm the the system earth-atmosphere, as suggested in Tosca, M.G. et al.. Attributing Accelerated Summertime Warming in the Southeast United States to Recent Reductions in Aerosol Burden: Indications from Vertically-Resolved Observations. Remote Sens. 2017, 9, 674.

The following sentences with two new references are now included in the Introduction."Clear-sky longwave forcing and cloudy-sky shortwave forcing of dust layer are very sensitive to the layer altitude, while the sign of the radiative effect of a biomass burning smoke layer depends on the presence of underlying stratus (Mishra et al. 2015, Tosca et al. 2017)."

Line 17 Pag 3: actually the adopted method in MPLNET retrievals pairing lidar measurements with sun-photometer data is used firstly by Marenco et al., 1997

The reference to Marenco et a:. 1997 has been made in the introduction.

Line 24 Pag. 4 How much "cloudy" should be a profile to be deleted? Please specify

The selection of "clear sky periods" is based on the removal of 1-min vertical profiles with cloud layers below 5 km, before looking at 30-min period with clear sky periods. The few lines describing the data selection now reads: "Data filtering with lidar radiometric detection of a cloud (day only), with search for layers showing very strong backscatter below 5 km (day and night) and for high opacity of the 0-3 km atmospheric layer (day and night), selects very efficiently the 30-min time periods with no cloud layers below 5 km. The following criteria are then applied to eliminate the lidar data considered too cloudy: all the profiles with a daytime sky level (SB) greater than 7000 counts/s or with a 150 m layer where the backscatter ratio is greater than 17 between 0 and 4.5 km, or with attenuated backscatter smaller than $10^{-4}$ km$^{-1}$sr$^{-1}$ between 3 and 8 km cloud layers below 5 km".